EMBO
Molecular Medicine

# P2X4 receptor controls microglia activation and favors remyelination in autoimmune encephalitis

Alazne Zabala[1], Nuria Vazquez-Villoldo[1], Björn Rissiek[2], Jon Gejo[1], Abraham Martin[3], Aitor Palomino[1], Alberto Perez-Samartín[1], Krishna R Pulagam[3], Marco Lukowiak[2], Estibaliz Capetillo-Zarate[1,4], Jordi Llop[3], Tim Magnus[2], Friedrich Koch-Nolte[5], Francois Rassendren[6], Carlos Matute[1,*] & María Domercq[1,**]

## Abstract

Microglia survey the brain microenvironment for signals of injury or infection and are essential for the initiation and resolution of pathogen- or tissue damage-induced inflammation. Understanding the mechanism of microglia responses during pathology is hence vital to promote regenerative responses. Here, we analyzed the role of purinergic receptor P2X4 (P2X4R) in microglia/macrophages during autoimmune inflammation. Blockade of P2X4R signaling exacerbated clinical signs in the experimental autoimmune encephalomyelitis (EAE) model and also favored microglia activation to a pro-inflammatory phenotype and inhibited myelin phagocytosis. Moreover, P2X4R blockade in microglia halted oligodendrocyte differentiation *in vitro* and remyelination after lysolecithin-induced demyelination. Conversely, potentiation of P2X4R signaling by the allosteric modulator ivermectin (IVM) favored a switch in microglia to an anti-inflammatory phenotype, potentiated myelin phagocytosis, promoted the remyelination response, and ameliorated clinical signs of EAE. Our results provide evidence that P2X4Rs modulate microglia/macrophage inflammatory responses and identify IVM as a potential candidate among currently used drugs to promote the repair of myelin damage.

**Keywords** microglia; myelin phagocytosis; P2X4 receptor; remyelination
**Subject Categories** Immunology; Neuroscience; Pharmacology & Drug Discovery

See also: **F Di Virgilio & AC Sarti** (August 2018)

## Introduction

Multiple sclerosis (MS) is a chronic inflammatory disease of the brain and spinal cord leading to demyelination and neurodegeneration. The clinical disease course usually starts with reversible episodes of neurological disability (relapsing–remitting MS; RRMS), which later goes into a progressive stage with irreversible neurological decline (secondary progressive MS; SPMS; Dendrou *et al*, 2015). Demyelinated lesions, a hallmark of multiple sclerosis, are caused by immune cell infiltration across the blood–brain barrier (BBB) that promotes inflammation, demyelination, gliosis, and neuroaxonal degeneration (Dendrou *et al*, 2015; Lassmann & Bradl, 2017). Axonal loss occurs in both the acute and chronic phases of MS and its animal model experimental autoimmune encephalomyelitis (EAE), and the loss of compensatory central nervous system (CNS) mechanisms contributes to the transition from RRMS to SPMS (Ransohoff, 2012; Dendrou *et al*, 2015). Activated microglia and macrophages are thought to contribute to neurodegeneration as their number correlates with the extent of axonal damage in MS lesions (Bitsch *et al*, 2000; Rasmussen *et al*, 2007; Fischer *et al*, 2013; Vogel *et al*, 2013). The activation of microglia/macrophages may represent one of the initial steps in EAE pathogenesis, preceding and possibly triggering T-cell development and infiltration of blood-derived cells (Heppner *et al*, 2005; Ajami *et al*, 2011; Goldmann *et al*, 2013; Yamasaki *et al*, 2014; Yoshida *et al*, 2014). However, other studies indicate that microglia/macrophage activation counteracts pathological processes by providing neurotrophic and immunosuppressive factors and by promoting recovery (Kotter *et al*, 2006; Miron & Franklin, 2014; Lampron *et al*, 2015).

Microglia/macrophages are highly heterogeneous immune cells with a continuous spectrum of activation states (Xue *et al*, 2014). The so-called classically activated or pro-inflammatory and the alternatively activated or anti-inflammatory microglia/macrophages are

1 Achucarro Basque Center for Neurosciences, CIBERNED and Departamento de Neurociencias, Universidad del País Vasco, Leioa, Spain
2 Department of Neurology, University Medical Center, Hamburg, Germany
3 Molecular Imaging Unit, CIC biomaGUNE, San Sebastian, Spain
4 IKERBASQUE, Basque Foundation for Science, Bilbao, Spain
5 Institute of Immunology, University Medical Center, Hamburg, Germany
6 Institut de Génomique Fonctionnelle, CNRS UMR5203, Montpellier, France
*Corresponding author. Tel: +34 94 6013244; Fax: +34 94 6015055; E-mail: carlos.matute@ehu.es
**Corresponding author. Tel: +34 94 6015681; Fax: +34 94 6015055; E-mail: maria.domercq@ehu.es

at the opposite ends of this spectrum (Mosser & Edwards, 2008; Murray et al, 2014; but see also Ransohoff, 2016). Although such a classification underestimates the complexity of macrophage/microglia plasticity, the distinction nevertheless provides a useful framework for exploring the diverse functions of the innate immune system in disease pathogenesis. Anti-inflammatory macrophages have been shown to play central roles in mediating Th2 immunity, wound healing, and the suppression of effector T-cell function (Mosser & Edwards, 2008). In MS, pro-inflammatory microglia exist in all types of lesions and correlate with axonal damage, whereas anti-inflammatory microglia are increased in acute active lesions and in the rim of chronic active lesions where efficient remyelination occurs (Miron et al, 2013). Anti-inflammatory microglia secrete anti-inflammatory cytokines and growth factors that promote oligodendrocyte progenitor differentiation and that protect neurons from damage (Butovsky et al, 2006; Mikita et al, 2011; Starossom et al, 2012; Miron et al, 2013; Yu et al, 2015). Finally, a block in the pro-inflammatory-to-anti-inflammatory switch has been hypothesized to contribute to remyelination failure in chronic inactive MS lesions (Miron et al, 2013; Sun et al, 2017).

As key immune effector cells of the CNS, surveillant microglia act as sensor of infection and pathologic damage of the brain, leading to a rapid plastic process of activation that culminates in the endocytosis and phagocytosis of damaged tissue. Multiple signals converge on microglial cells to actively maintain or alter their functional state and orchestrate the specific repertoire of microglial functions. In the absence of pathogens, microglia sense the injury by recognizing the release of molecules that are normally located inside the cell, known as damage-associated molecular patterns (DAMPs) or "endogenous danger signals" (Di Virgilio, 2007). Recently, ATP has been characterized as a danger signal implicated in innate and adaptive immunity (Junger, 2011), leading to a plethora of responses in microglia through its interaction with their purinergic P2 receptors (Domercq et al, 2013). On the basis of their signaling properties, P2 receptors can be further subdivided into metabotropic P2Y receptors (P2YRs) that are G-protein-coupled, and ionotropic P2X receptors (P2XRs) that are nucleotide-gated ion channels (Domercq et al, 2013). We have previously observed that purinergic P2X4R is highly expressed in activated microglia in EAE and in human MS optic nerve samples (Vázquez-Villoldo et al, 2014). Here, we identified P2X4R as a significant regulator of microglia inflammatory cascade and the resultant repair response after demyelination.

# Results

## P2X4R expression is upregulated during EAE

Following peripheral nerve injury, microglia in the spinal dorsal horn exhibit a reactive phenotype and upregulate expression of a variety of genes, including purinergic P2x4r (Tsuda et al, 2003; Beggs et al, 2012). Accordingly, we detected previously an increase in P2x4r mRNA expression in multiple sclerosis (MS) samples and at the peak of the immune attack in the acute EAE model (Vázquez-Villoldo et al, 2014). We have further analyzed the time course of the P2x4r expression in EAE mice immunized with myelin oligodendrocyte glycoprotein (MOG). Levels of P2x4r expression were increased at the peak of the disease and remained elevated during the recovery phase (30 days; Fig 1A). Interestingly, there was a strong correlation between P2x4r expression and the neurological score, both at the peak and at recovery (Fig 1A; $r^2$ = 0.99 and 0.61, respectively). Previous data have demonstrated that interferon regulatory factor 8 (IRF8)–IRF5 transcriptional axis is a critical regulator for shifting microglia toward a P2X4R+-reactive phenotype (Masuda et al, 2014). Accordingly, we observed that Irf8 and Irf5 transcription factors were upregulated at the peak and recovery phases of the disease and their expression correlated well with P2x4r expression (Fig 1B). P2x4r upregulation was also detected in FACS-isolated microglia (Cd11b+CD45high) in the spinal cord at the EAE recovery phase (Fig 1C).

## P2X4R blockade exacerbates EAE

We then tested the role of P2X4R in EAE pathogenesis in mice treated daily with P2X4R antagonist TNP-ATP (10 mg/kg) from the onset of the disease at 10 days postimmunization (dpi). This time window is coincident with microglia activation, as previously reported (Ajami et al, 2011), and does not interfere with immune priming. Microglia die at early stages of EAE induction, and this population is replenished by infiltrating monocytes, promoting progression to paralysis (Ajami et al, 2011). Because P2X4R blockade was previously demonstrated to prevent LPS-induced microglial cell death (Vázquez-Villoldo et al, 2014), we reasoned that blockade of microglial cell death by TNP-ATP would prevent replacement by monocyte and improve clinical signs of EAE. In contrast, blockade of P2X4R with TNP-ATP exacerbated EAE disease (Fig 2A). Accordingly, the latency of the corticospinal tract was significantly increased in TNP-ATP-treated mice (Fig 2A), indicative of higher demyelination. We next stained spinal cord sections at the end of the experiment for ionized calcium-binding adapter protein 1 (Iba1), which is a marker commonly used to identify microglia/macrophages in the CNS. TNP-ATP-treated mice showed a significant increase in Iba1+ cells in the white and gray matter of the spinal cord versus vehicle-treated EAE mice (Fig 2B). The increase was observed even at the same neurological score (Fig EV1), probably indicating that the increase in microglia/macrophage number is not only the consequence of the higher EAE severity after P2X4R blockade.

We next confirmed the role played by P2X4R in EAE pathogenesis using P2X4−/− mice. We first checked whether P2X4R deficiency could affect microglia and oligodendrocytes in normal conditions. We did not detect any change in the number or morphology of Iba1+ cells nor in the number of Olig2+ oligodendrocytes in the spinal cord of 2-month-old P2X4−/− mice (Appendix Fig S1). Then, we compared neurological score in WT and P2X4−/− MOG-injected mice. In accordance with results obtained with TNP-ATP, P2X4−/− mice showed an exacerbated EAE, higher latency of the corticospinal tract, and an increase in the number of Iba1+ microglia/macrophage cells (Fig 2C). To further assess that the effect of TNP-ATP was P2X4R-dependent, we treated P2X4−/− mice with TNP-ATP from the onset of the disease. TNP-ATP failed to alter the course of EAE disease in P2X4−/− mice (Fig 2D). All these data confirmed the role played by P2X4R in EAE pathogenesis.

During EAE, T cells are primed in the mouse peripheral immune system before the onset of the clinical signs (Stromnes & Goverman,

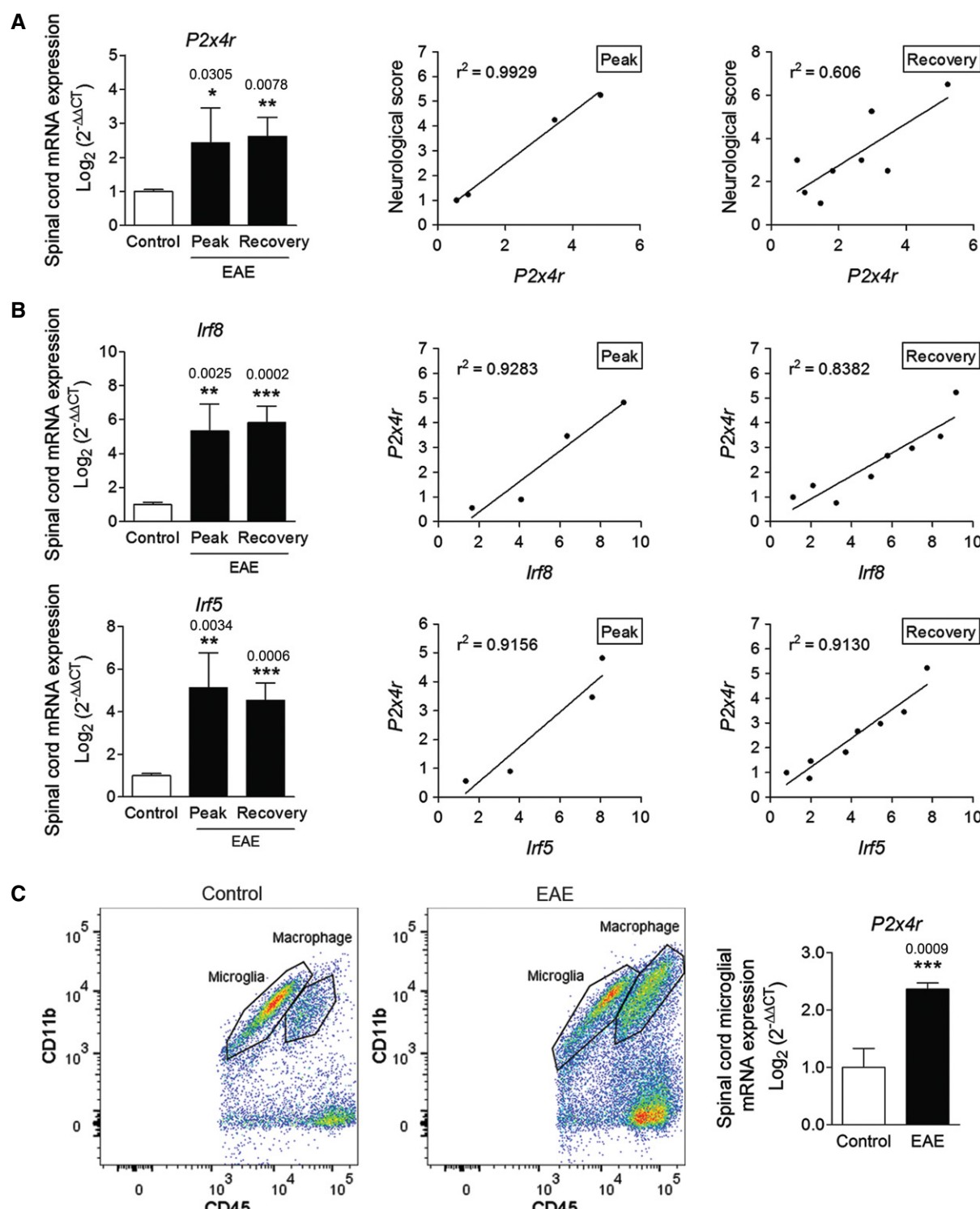

**Figure 1. P2x4r is increased at the peak and recovery phases of MOG-EAE mice.**

A   Expression of *P2x4r* in the spinal cord of control (*n* = 7) and of EAE mice at the peak (*n* = 4) and recovery (*n* = 8) phases as analyzed using qPCR. *Right*, correlation of *P2x4r* expression with neurological score at EAE peak and recovery.

B   Expression of *Irf8* and *Irf5* in the spinal cord of control (*n* = 7) and of EAE mice at the peak (*n* = 4) and recovery (*n* = 8) phases. *Right*, correlation of *P2x4r* expression with *Irf5* and *Irf8* expression at EAE peak and recovery.

C   Plots depicting the strategy to distinguish resident microglia (CD11b[+]/CD45[low]) from invading macrophages (CD11b[+]/CD45[high]) in naïve and EAE mouse spinal cords. *Right*, *P2x4r* expression in microglia from naïve mice (*n* = 4) and from EAE mice at the recovery phase (*n* = 6).

Data information: Data are presented as mean ± s.e.m. and were analyzed by one-way ANOVA (A, B) and Student's *t*-test (C). *$P < 0.05$, **$P < 0.01$, ***$P < 0.001$.

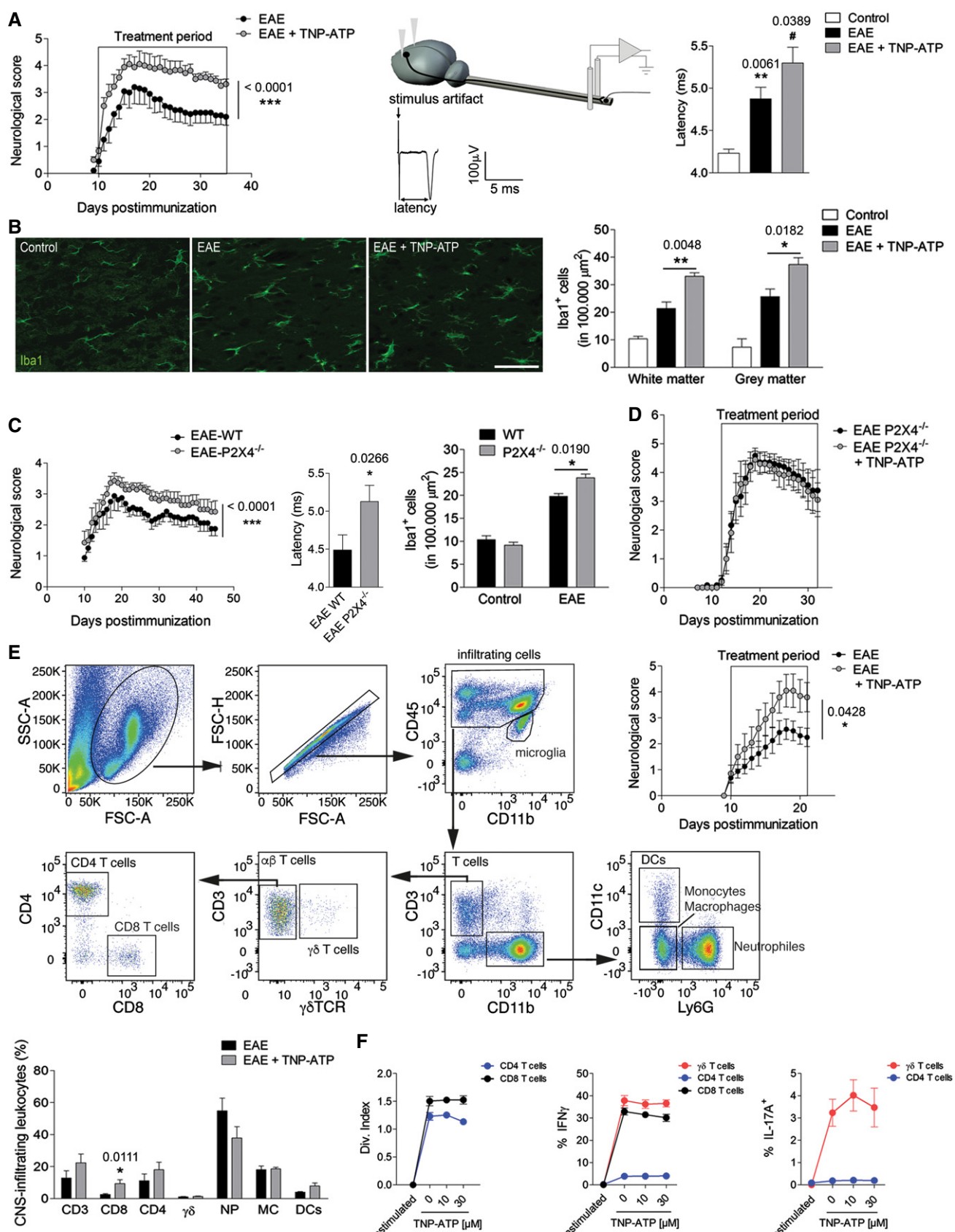

Figure 2.

**Figure 2. P2X4R blockade increases autoimmune inflammation.**

A Clinical score of vehicle ($n = 10$)- and TNP-ATP (10 mg/kg)-treated mice ($n = 10$) after EAE induction. *Right*, scheme, raw data, and histogram showing axon conduction latency in the corticospinal tract of control ($n = 4$), EAE ($n = 10$), and TNP-ATP-treated EAE mice ($n = 10$). Symbols indicate significance versus control (*) or versus EAE ([#]).

B Histology of spinal cord sections using Iba1 antibodies shows a significant increase in microglia/macrophage cell number in control mice ($n = 3$), in TNP-ATP-treated EAE mice ($n = 4$), and in non-treated EAE mice ($n = 5$). Scale bar = 50 μm.

C Neurological score, axon conduction latency, and microglia/macrophage quantification in control WT ($n = 3$) and P2X4$^{-/-}$ mice ($n = 3$) and after EAE induction in WT ($n = 10$) and P2X4$^{-/-}$ mice ($n = 10$).

D Neurological score after EAE induction in P2X4$^{-/-}$ mice treated with vehicle ($n = 4$) or TNP-ATP ($n = 5$).

E *Left*, flow cytometry gating strategy for analysis of infiltrates in the spinal cord of EAE mice at peak (18–21 dpi). *Right*, graph showing the neurological score of the mice used for the analysis (top) ($n = 8$). Histogram showing flow cytometric quantification of CD3$^+$, CD8$^+$, CD4$^+$, and γδ T cells, neutrophils (NP), macrophages (MC), and dendritic cells (DC) in the spinal cord at EAE peak (bottom).

F Immune response analysis *in vitro*. T-cell proliferation assay and flow cytometric quantification of cytokine expression in T cells after PMA/ionomycin stimulation in the absence or presence of TNP-ATP (10–30 μm, 3 days; $n = 3$ independent experiments).

Data information: Data are presented as mean ± s.e.m. Statistics were performed with Mann–Whitney $U$-test (neurological score) and Student's $t$-test. *$^{/#}P < 0.05$, **$P < 0.01$, ***$P < 0.001$.

2006), suggesting that treatment with TNP-ATP starting after the onset of the disease would have no impact on T-cell infiltration. In order to corroborate this hypothesis, we analyzed immune cell infiltration (CD4$^+$ T cells, CD8$^+$ T cells, γδ T cells, neutrophils, and macrophages) in the brain (Fig EV1) and spinal cord (Fig 2E) at the peak of disease in vehicle- and TNP-ATP-treated mice by flow cytometry in another set of mice. We found similar proportions for all of CNS-infiltrating CD45$^+$ leukocytes in TNP-ATP- and vehicle-treated mice at the peak of EAE except for CD8$^+$ T cells in the spinal cord (Fig 2E) and brain (Fig EV1) which were significantly increased in TNP-ATP-treated mice. We further assessed the role of P2X4R on T-cell function directly by stimulating isolated splenocytes *in vitro* with anti-CD3/CD28 in the presence or absence of TNP-ATP for 3 days and then measured proliferation of CD4$^+$ and CD8$^+$ T cells. Furthermore, we determined T-cell cytokine production via intracellular cytokine staining of PMA/ionomycin-stimulated cells. Importantly, *in vitro* assays showed that blockade of P2X4R on lymphocytes did not alter T-cell proliferation or cytokine production (Fig 2F).

To further exclude the involvement of P2X4R in adaptive immune system, we performed an additional EAE experiment and treated mice with TNP-ATP during the priming phase (0–17 dpi). TNP-ATP treatment did not affect disease development (Fig 3A). At the peak, we quantified by flow cytometry the immune response in periphery (spleen and lymph nodes) and in the spinal cord. Treatment with TNP-ATP during the priming phase did not change the number of CD4$^+$ T cells, CD8$^+$ T cells, and γδ T cells in spleen, lymph nodes, or the spinal cord (Fig 3B). To further assess the CD4$^+$ T-cell response, we measured mRNA for *Foxp3* and *Ror*, transcription factors that specify Tregs and Th17, respectively, and *Ifng*, signature cytokine for Th1 cells. We did not detect any change in transcript expression after TNP-ATP treatment (Fig 3C). Finally, to check whether P2X4R blockage could alter infiltration of T cells during the immune priming phase, we measured BBB disruption by PET imaging using a radioligand that enabled tracking of matrix metalloproteinase (MMP) activity as a marker of early lesions and ongoing leukocyte infiltration (Gerwien *et al*, 2016). An increase in MMP activity was detected in the lumbar spinal cord at the peak of the disease; however, treatment with TNP-ATP during the immune priming did not change the MMP-PET signal in EAE (Fig 3D). Altogether, these data suggest that blockade of P2X4R did not interfere with the efficacy of immunization and the T-cellular immune response against MOG.

## Role of P2X4R on microglia polarization

Since myelin clearance is necessary for remyelination and recovery (Li *et al*, 2005; Kotter *et al*, 2006; Neumann *et al*, 2009) and phagocytosis and remyelination are modulated by microglia/macrophage polarization (Miron *et al*, 2013), we hypothesized that P2X4R could be involved in this process. We first checked the status of microglia/macrophage in TNP-ATP- and vehicle-treated EAE mice. We performed gene expression profiling from the lumbar spinal cord of vehicle- and TNP-ATP-treated EAE mice at the peak and recovery phases (Fig 4A and B). Expression of pro-inflammatory and anti-inflammatory genes involved in microglia/macrophage activation was analyzed using a 96.96 Dynamic Array™ integrated fluidic circuit (Fluidigm). In accordance with previous data showing that macrophages and microglia showed an intermediate activation status in MS (Vogel *et al*, 2013), most pro-inflammatory and anti-inflammatory genes were significantly increased at the EAE peak (Fig EV2) and recovery phase (Fig 4B). Blockade of P2X4R with TNP-ATP did not significantly alter anti-inflammatory gene expression, but it significantly increased pro-inflammatory gene expression at the recovery phase (Fig 4B), but not at EAE peak (Fig EV2). A higher increase in pro-inflammatory gene expression was also detected on microglia FACS-sorted (Cd11b$^+$CD45$^{high}$; see gating in Fig 1C) from EAE P2X4$^{-/-}$ mice versus EAE WT (Fig 4C). Accordingly, we found an increase in iNOS expression in microglia/macrophage after EAE in P2X4$^{-/-}$ mice and in TNP-ATP-treated mice (Fig 4D and E). These data suggest that P2X4R could be influencing microglia/macrophage activation.

To further analyze the influence of P2X4R on microglia polarization *in vitro*, cells were primed with colony-stimulating factors to differentiate into pro-inflammatory and anti-inflammatory microglia according to a previous protocol (Fig 5A; see details in Materials and Methods) and analyzed by immunocytochemistry using pro-inflammatory (iNOS) and anti-inflammatory (mannose receptor; MRC1) markers. Blockade of P2X4R with TNP-ATP induced a significant increase in iNOS$^+$ cells and a significant reduction in MRC1$^+$ cells (Fig 5B). Accordingly, qPCR analysis revealed an increase in pro-inflammatory genes and a decrease in anti-inflammatory genes after TNP-ATP treatment during polarization (Fig 5C). Similar results were obtained in P2X4$^{-/-}$ microglia (Fig EV3). Altogether, these data suggest that P2X4Rs modulate microglial polarization.

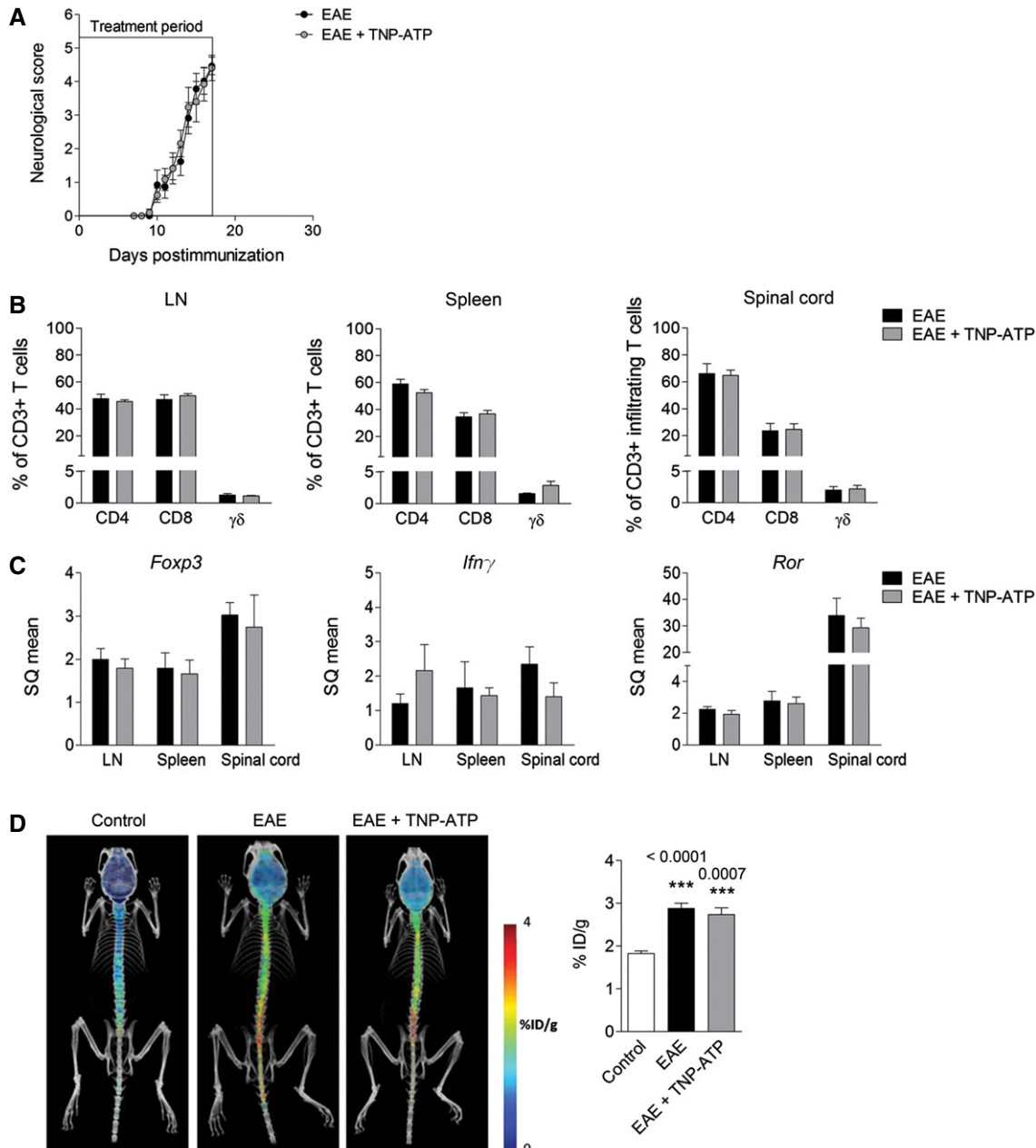

**Figure 3. P2X4R does not interfere with immune priming.**

A   Neurological score of vehicle (n = 23 from three independent experiments)- and TNP-ATP-treated (n = 21 from three independent experiments) mice after EAE. Mice were treated daily with TNP-ATP from 0 dpi to EAE peak.

B   Flow cytometric quantification of CD8⁺, CD4⁺, and γδ T cells in lymph nodes (LN), spleen, and spinal cord at EAE peak of vehicle (n = 7)- and TNP-ATP-treated mice (n = 5) (gating strategy as described in Fig 2E).

C   Relative mRNA expression of *Foxp3*, *Ifng*, and *Ror* in LN, spleen, and spinal cord at EAE peak of vehicle (n = 6)- and TNP-ATP-treated mice (n = 7).

D   Representative images of ¹⁸F-MMPi PET imaging in control mice (n = 6), EAE mice (n = 10), and EAE mice treated with TNP-ATP (n = 9) as described in (A). ¹⁸F-MMPi signal in the lumbar spinal cord was expressed as %ID/g.

Data information: Data are presented as mean ± s.e.m. and were analyzed by one-way ANOVA. ***P < 0.001.

## Effect of P2X4R on oligodendrocyte differentiation and myelination

A switch from a pro-inflammatory to an anti-inflammatory phenotype occurs in microglia and peripherally derived macrophages during remyelination in MS, and this change is essential for efficient remyelination (Miron *et al*, 2013). These data led us to the hypothesis that P2X4R blockade in microglia could be indirectly affecting oligodendrocyte differentiation and remyelination. We first characterized the expression and function of P2X4R in oligodendrocytes

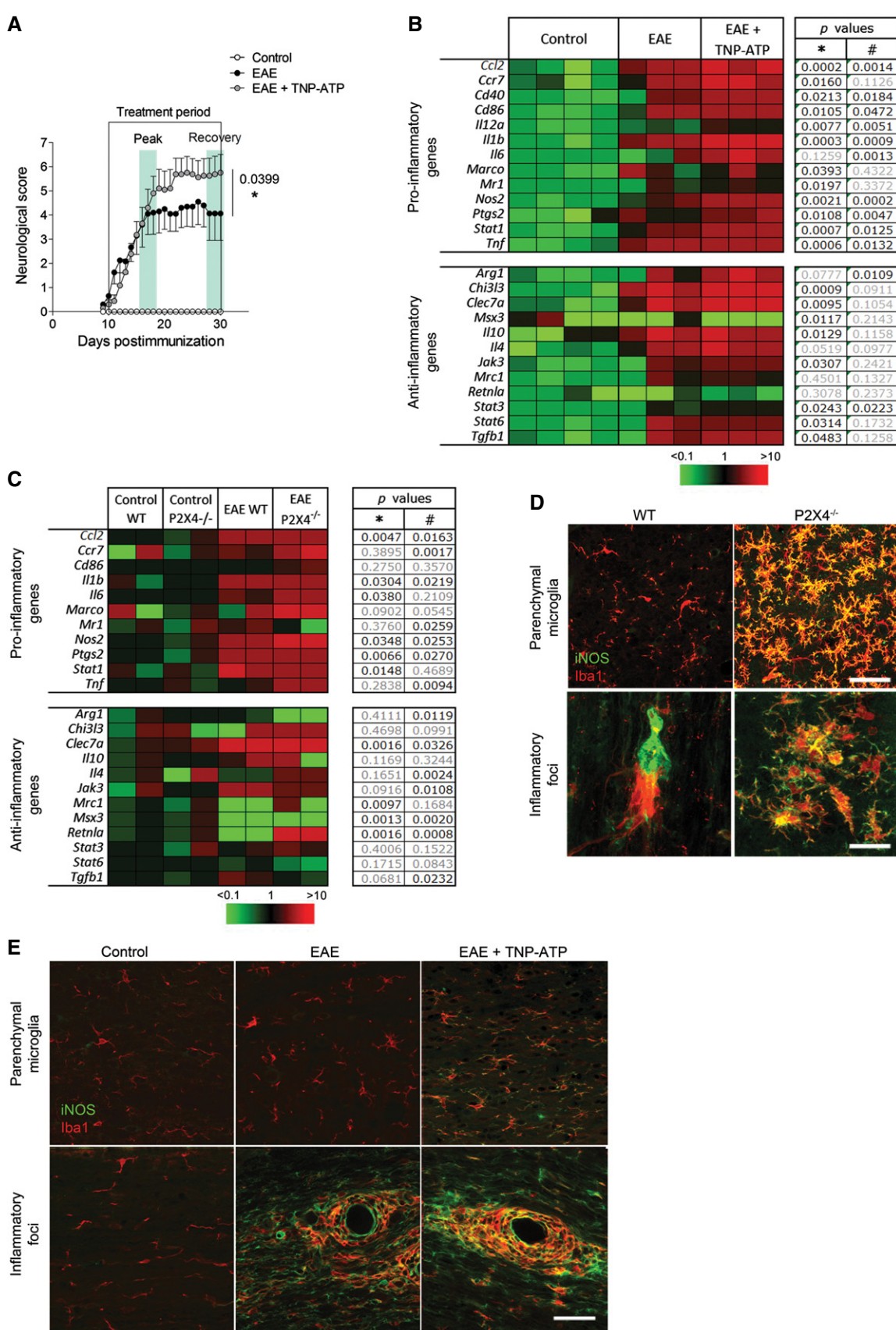

**Figure 4.**

◄

**Figure 4. P2X4R blockade increases pro-inflammatory gene expression after EAE.**

A   Neurological score of EAE (*n* = 4) or TNP-ATP-treated EAE mice (*n* = 4) used for gene expression analysis. Mice were treated from the onset to the end of the experiment. Data are presented as mean ± s.e.m. and were analyzed by Mann–Whitney *U*-test. *P < 0.05.

B   Heatmap showing significant changes in pro-inflammatory and anti-inflammatory mRNA expression in the spinal cord at the EAE recovery phase in the presence or absence of TNP-ATP (*n* = 3). The expression levels of genes are presented using fold-change values transformed to Log2 format compared to control. The Log2 (fold-change values) and the color scale are shown at the bottom of the heatmap. Tables indicate statistical significance between control and EAE (*) and between EAE and EAE + TNP-ATP ($^{#}$). Data were analyzed by Student's *t*-test.

C   Heatmap showing changes in pro-inflammatory and anti-inflammatory markers in FACS-isolated microglia from control, EAE WT, and EAE P2X4$^{-/-}$ mice at the recovery phase (*n* = 2 in duplicate). Tables indicate statistical significance between control WT and EAE WT (*) and between EAE WT and EAE P2X4$^{-/-}$ ($^{#}$). Data were analyzed by Student's *t*-test.

D, E  iNOS (green) expression was increased in Iba1$^{+}$ cells (red) in the spinal cord of P2X4$^{-/-}$ mice versus WT mice (D) and in TNP-ATP-treated mice versus non-treated mice after EAE induction (E) (*n* = 3). Analysis was performed at the recovery phase of the EAE. Scale bar = 50 (D, top; E) and 25 (D, bottom) μm.

and microglia. Double-immunocytochemistry analysis showed that P2X4R expression is virtually absent from Olig2$^{+}$ oligodendrocytes and highly enriched in isolectin B4$^{+}$ microglial cells in microglia–oligodendrocyte progenitor cell (OPC) mixed cultures (Fig 6A; Appendix Fig S2). On the contrary, P2X7 receptors are expressed in both cell populations. The function of P2X4R in microglia and oligodendrocytes was further analyzed by electrophysiology in acute slices from CXCR3-GFP and PLP-DsRed mice, respectively. When ATP was applied to whole-cell-clamped microglia at −70 mV, it elicited a massive inward current (619 ± 100 pA; *n* = 8) that was significantly reduced by A438079 (10 μM), antagonist of P2X7R (172 ± 69 pA; *n* = 8; Fig 6B). However, the remaining ATP-evoked current was virtually abolished in the presence of TNP-ATP (Fig 6B). These data suggest that P2X7R and P2X4R are the main contributors to ATP-evoked currents in microglia. In contrast, ATP-evoked currents in oligodendrocytes (80 ± 21 pA) were totally abolished in the presence of A438079 (Fig 6B), indicating the lack of functional P2X4R on these cells. To further exclude any direct role of P2X4R on oligodendrocyte differentiation, we stimulated culture oligodendrocytes with ATPγS (10 μM, 3 days) at low concentrations to avoid P2X7R activation. ATPγS did not induce any change in oligodendrocyte differentiation (Fig EV4). Altogether, these results suggested that oligodendrocytes lack functional P2X4Rs.

Next, we analyzed whether microglia P2X4R could play a role on oligodendrocyte differentiation. To this end, we conditioned oligodendrocyte culture medium (SATO; see Materials and Methods) with control, pro-inflammatory, and anti-inflammatory microglia for 24 h and then incubated oligodendrocyte progenitors with microglia-conditioned media for 3 days (see cartoon in Fig 6C). All microglia-conditioned media induced an increase in the number of mature MBP$^{+}$ oligodendrocytes as compared to polarization factors alone (Fig 6C). However, anti-inflammatory microglia induced a higher increase in oligodendrocyte differentiation, an effect that was blocked in the presence of TNP-ATP (Fig 6C). Since BDNF enhances oligodendrocyte differentiation and myelination (Wong *et al*, 2013) and P2X4R stimulation in microglia has been linked to BDNF release (Tsuda *et al*, 2003; Coull *et al*, 2005; see also Fig EV4), we analyzed BDNF production in differentially polarized microglia. Western blot analysis revealed a significant increase in BDNF production in anti-inflammatory microglia, which was significantly reduced in the presence of TNP-ATP (Fig 6D). Accordingly, *Bdnf* mRNA expression during the recovery phase of EAE was significantly reduced in the presence of TNP-ATP, a fact that correlated with the downregulation of *Mbp*

expression (Fig 6E). *Bdnf* mRNA was also reduced in FACS-isolated microglia from control and EAE P2X4$^{-/-}$ mice (Fig EV4).

These data suggested that P2X4R blockade in microglia inhibits oligodendrocyte differentiation by shifting microglia toward a pro-inflammatory phenotype, an effect that could impede remyelination. To assess that possibility, we further checked the impact of P2X4R on remyelination on the lysolecithin-induced demyelination model using *ex vivo* organotypic cerebellar slice culture (Fig 6F), a model independent of adaptive immune system. Indeed, slices treated with TNP-ATP (10 μM) during the remyelination phase (7 days) showed a significant decrease in their remyelination capacity, as revealed by analyzing MBP expression by Western blot (Fig 6F).

## Myelin clearance is modulated by P2X4Rs

Phagocytosis of myelin debris by microglia is essential for an efficient regenerative response (Kotter *et al*, 2006; Ruckh *et al*, 2012). We observed a higher accumulation of disrupted or fragmented myelin after EAE induction in TNP-ATP-treated mice (Fig 7A) as well as in P2X4$^{-/-}$ mice (Fig EV5). Fragmented myelin yielded higher immunoreactivity likely due to additional exposed epitopes of the MBP antibody. Myelin debris accumulation could be the result of higher demyelination or a defect on myelin phagocytosis. Indeed, we observed higher accumulation of myelin debris not surrounded by microglia phagocytic processes in TNP-ATP-treated mice (Fig 7A) at the recovery phase. Because of that, we challenged the hypothesis that these features could be the consequence of a failure on microglia/macrophage phagocytosis. For that, myelin was isolated from adult rat whole brain using sucrose gradient (Norton & Poduslo, 1973), labeled with the dye Alexa 488, and added to microglia cultures. In order to efficiently clear up myelin, microglia should internalize myelin and deliver it to lysosomes to degrade it. We monitored by confocal microscopy myelin endocytosis and myelin degradation time course. At initial stages, myelin endocytosis (1 h) was significantly increased in anti-inflammatory microglia (Fig 7B), in accordance with recent data (Healy *et al*, 2016). Blockade of P2X4R with TNP-ATP significantly reduced the anti-inflammatory-dependent increase in myelin endocytosis (Fig 7B). We then checked myelin degradation in the different polarized microglia populations. After 1-h incubation with Alexa 488-myelin, microglia were kept in label-free growth medium for up to 6 days in the presence of polarizing factors. We could not eliminate polarization factors because microglia activation quickly reverses without them (unpublished observations). We found that microglia retained Alexa 488-labeled myelin for up to 6 days and only about 30% of

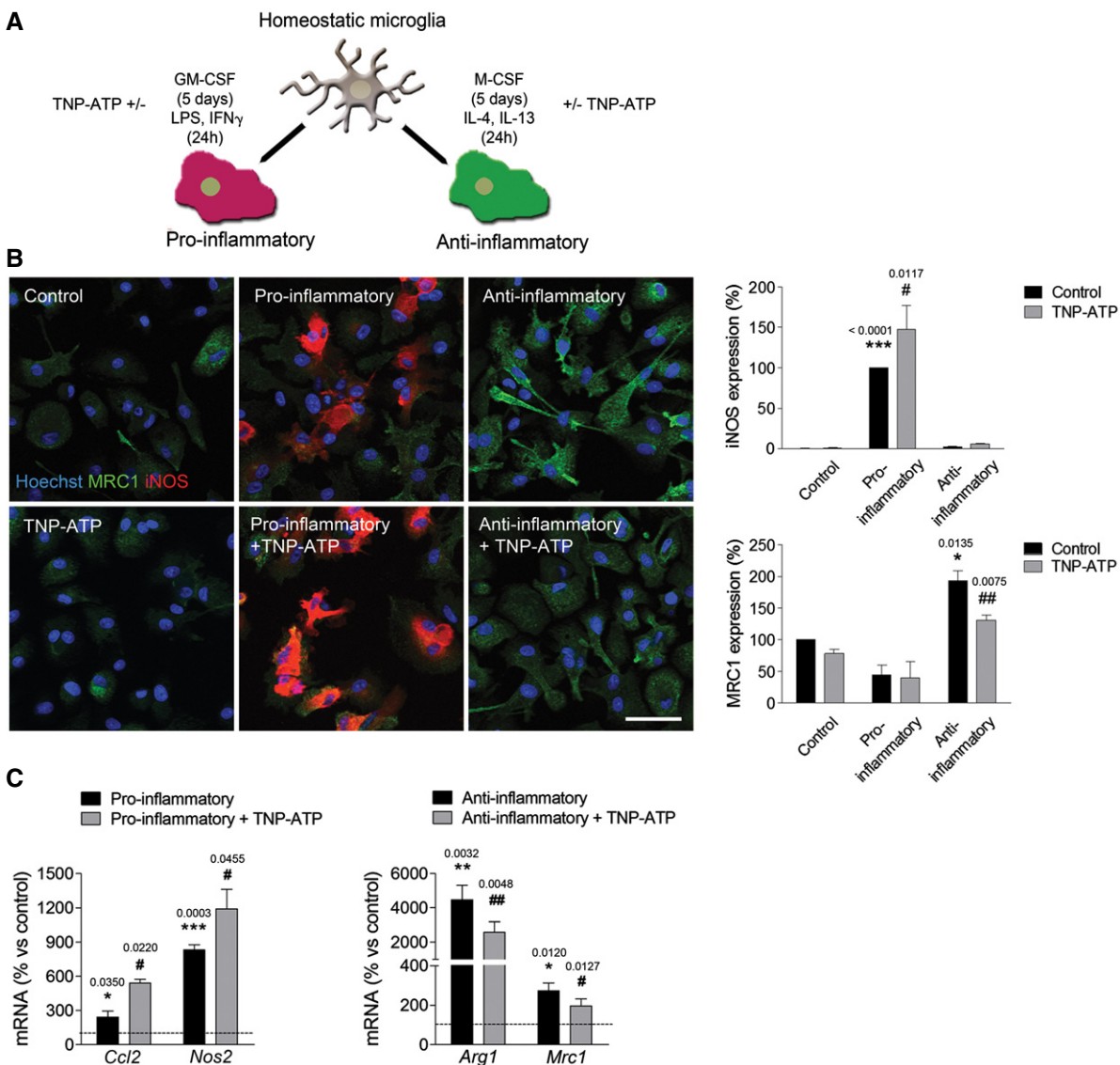

**Figure 5.  P2X4R modulates microglia polarization.**

A   Schematic representation of microglia polarization protocol.
B   Staining for iNOS (red) (*n* = 7) and mannose receptor (MRC1, green) (*n* = 4) in different activated microglia in the absence or presence of TNP-ATP (10 μM). Scale bar = 50 μm.
C   qPCR quantification of pro-inflammatory genes (*Ccl2* and *Nos2*) and anti-inflammatory genes (*Arg1* and *Mrc1*) in different activated microglia in the absence or presence of TNP-ATP (10 μM) (*n* = 3).

Data information: Data are presented as mean ± s.e.m. and were analyzed by one-way ANOVA. \*/\#*P* < 0.05, \*\*/\#\#*P* < 0.01, \*\*\**P* < 0.001 versus control (\*) or versus pro-/anti-inflammatory microglia (\#).

internalized myelin was degraded in that time in control microglia (Fig 7C). Anti-inflammatory microglia showed a higher capacity to degrade myelin at 3 and 6 days, whereas pro-inflammatory microglia degraded myelin less efficiently (Fig 7C). Moreover, P2X4R blockade by TNP-ATP significantly attenuated the increase on myelin degradation observed in anti-inflammatory microglia and exacerbated the deficits on myelin degradation characteristic of pro-inflammatory microglia (Fig 7D). These results suggest that P2X4R blockade could contribute to the failure on myelin phagocytosis necessary prior to regeneration.

**P2X4R potentiation ameliorates EAE**

We then explored the therapeutic potential of P2X4R potentiation. Ivermectin (IVM) is an FDA-approved semi-synthetic macrocyclic lactone used in veterinary and clinical medicine as an anti-parasitic agent. IVM allosterically modulates both ion conduction and channel gating of P2X4Rs (Priel & Silberberg, 2004). We first analyzed the role of P2X4R potentiation in EAE pathogenesis in mice treated daily with IVM (1 mg/kg) after the onset of the disease (14 dpi). IVM induced a rapid and significant amelioration of the motor

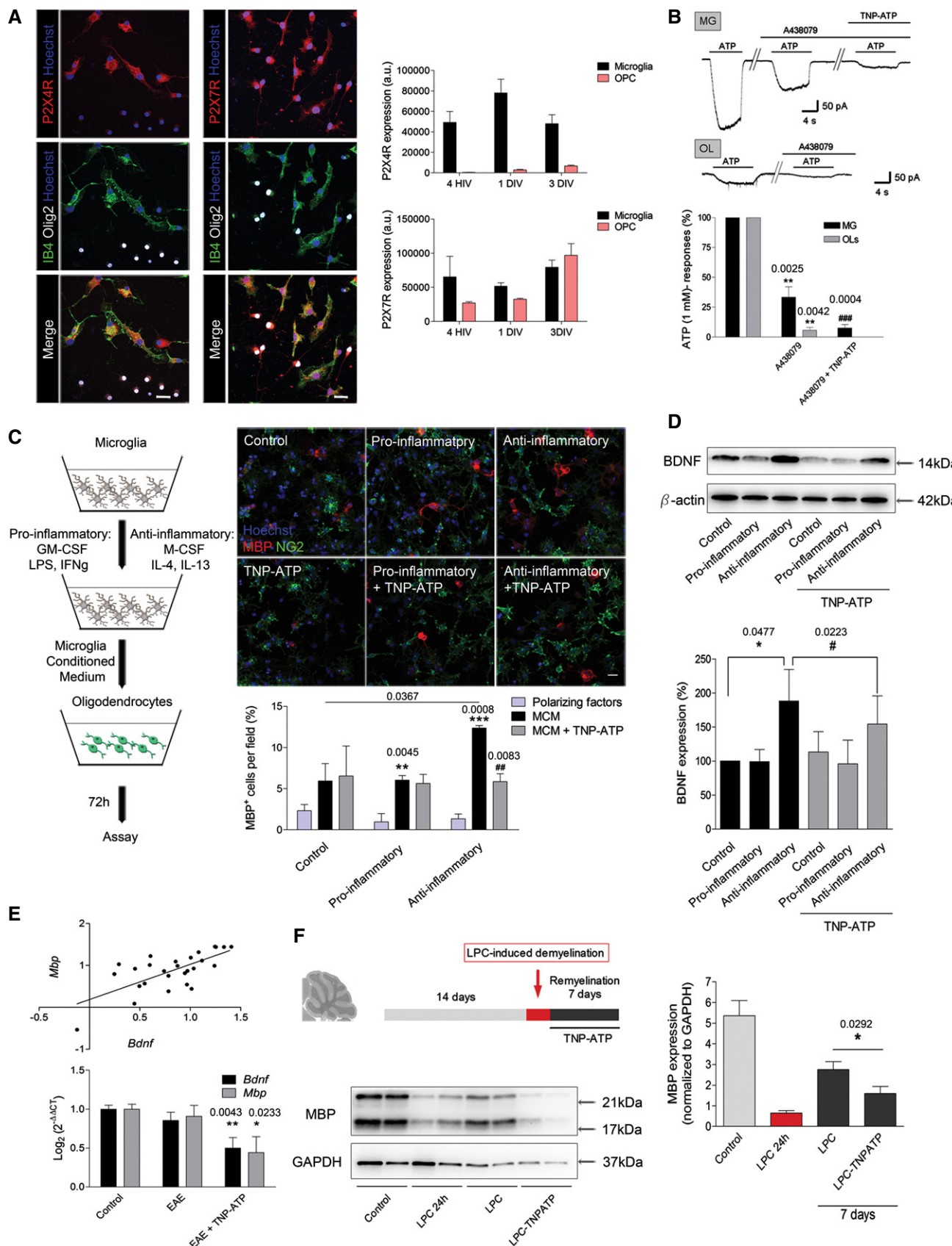

Figure 6.

◀

**Figure 6.  Microglial P2X4R modulates oligodendrocyte differentiation.**

A  Microglia–OPC coculture stained for P2X4R (left column) or P2X7R (right column) (red), Olig2 (white), isolectin B4 (IB4; green), and Hoechst (blue). Scale bar = 20 μm. Notice the presence of P2X4R on isolectin B4$^+$ microglial cells and the absence in Olig2$^+$ cells. Histograms *(right)* show quantification of P2X4R and P2X7R at different stages of oligodendrocyte development, 4 h *in vitro* (HIV), 1 day *in vitro* (DIV), and 3 DIV in cocultures (n = 3).

B  Microglia (MG) express functional P2X7R (blocked with A438079) and P2X4R (blocked with TNP-ATP), whereas oligodendrocytes (OL) only express functional P2X7 receptors. Representative traces showing ATP (100 μM)-evoked inward, non-desensitizing currents in microglia and oligodendrocytes in acute slices in the absence or presence of P2X7R antagonist A438079 (1 μM) or of P2X4R antagonist TNP-ATP (10 μM) (n = 9 cells from three different mice). Symbols indicate significance versus ATP currents (*) or versus ATP+A438079 currents ($^{\#}$).

C  OPCs were treated with control, anti-inflammatory, and pro-inflammatory microglia-conditioned media (MCM) or with fresh media ± polarizing factors and stained for NG2 (green) and MBP (red). Scale bar = 20 μm (n = 3). Symbols indicate significance versus polarizing factors (*) or versus MCM ($^{\#}$).

D  Representative immunoblots of BDNF levels in microglia *in vitro* and densitometry quantification (n = 10).

E  *Bdnf* and *Mbp* mRNA expression (bottom) and correlation of the expression (top) at the EAE recovery phase in control mice, EAE mice, and TNP-ATP-treated EAE mice (n = 10 mice/group).

F  Sagittal sections of cerebellum (300 μm) were treated with lysolecithin (LPC) to induce demyelination and allowed to remyelinate ± TNP-ATP (10 μM) during 7 days. Representative immunoblots of MBP and GAPDH and densitometry quantification (n = 5).

Data information: Data are presented as mean ± s.e.m. and were analyzed by one-way ANOVA (B, C, E) and Student's *t*-test (D, F). *$^{/\#}P$ < 0.05, **$^{/\#\#}P$ < 0.01, ***$^{/\#\#\#}P$ < 0.001.

Source data are available online for this figure.

deficits of EAE (Fig 8A) that lead to an improvement in corticospinal tract function in the recovery phase, as revealed by the decrease in the latency of corticospinal tract (Fig 8A). These data indicate that IVM treatment effectively promoted myelin recovery. Accordingly, IVM significantly enhanced remyelination in the lysolecithin-induced demyelination model in organotypic cerebellar slices (Fig 8B).

Then, we analyzed the effect of IVM on microglial polarization and phagocytosis *in vitro*. IVM (3 μM) induced a significant increase in polarization to an anti-inflammatory phenotype (Fig 8C). Accordingly, qPCR analysis revealed a decrease in pro-inflammatory genes and an increase in anti-inflammatory genes after IVM treatment during polarization (Fig 8D). Finally, we analyzed the effect of IVM on myelin phagocytosis. We did not detect a potentiation of myelin endocytosis (1 h) in anti-inflammatory microglia, perhaps because myelin endocytosis capacity is already saturated. However, myelin endocytosis was significantly increased in pro-inflammatory microglia and even in control microglia in the presence of IVM (Fig 8E). Regarding myelin degradation, long-term treatment with ivermectin potentiated myelin degradation in control (Fig EV5), pro-inflammatory, and anti-inflammatory microglia (Fig 8F). The effect of IVM in myelin phagocytosis was absent in P2X4$^{-/-}$ microglia (Fig EV5). Previous studies have described P2X4R to be located intracellularly in late endosomes and lysosome membranes where they form functional ATP-activated cation channels regulated by luminal ATP in a pH-dependent manner (Huang *et al*, 2014). We observed that IVM induced lysosome fusion, an effect dependent on P2X4R (Fig EV5). Because fusion of lysosomes with late endosomes generates acidic endolysosomes with high cathepsin activity for phagocytic degradation, we then measured intralysosomal pH in microglia. Anti-inflammatory microglia showed an acidic shift of lysosomes, which correlates with its higher phagocytic capacity. Moreover, IVM treatment (3 μM, 16 h) induced a significant acidification of lysosomes (Fig EV5). Altogether, data suggested that P2X4R potentiation by IVM could favor microglia switch necessary to efficient remyelination.

The therapeutic potential of ivermectin in MS could be compromised by its possible influence in neuropathic pain. Thus, a recent study reported the anti-allodynic effect of a novel P2X4R antagonist in mice with traumatic nerve damage, although the antagonist did not affect acute nociceptive pain and motor function (Matsumura *et al*, 2016). We therefore measured the effects of IVM in mechanical allodynia at different stages of EAE before the appearance of severe motor deficits. Withdrawal thresholds were significantly diminished in both vehicle- and IVM-treated mice at EAE onset and before EAE peak relative to baseline responses, but there were no significant differences between the two cohorts at any stage (Fig 8G).

## Discussion

Brain injury induces an upregulation of P2X4R and shifts microglia toward a P2X4R-expressing reactive state through an IRF8–IRF5 transcriptional axis (Beggs *et al*, 2012). In the present study, we showed that IRF8–IRF5–P2X4R is upregulated in the peak and recovery phases of EAE. Moreover, we demonstrated that blockade of P2X4R exacerbates EAE, whereas potentiation with IVM ameliorates this experimental disease. Mechanistically, P2X4 receptor signaling potentiation in microglia/macrophages favors a switch to an anti-inflammatory phenotype that, by secreting factors such as BDNF and increasing myelin phagocytosis, leads to higher remyelination. Altogether, data here suggest that P2X4R upregulation could be a marker of the neuroinflammatory response in MS and that potentiation of signaling by P2X4R has therapeutic potential to treat demyelinating disorders.

Microglial P2X4R upregulation through IRF8–IRF5 transcription factors, the P2X4R$^+$ state of microglia, seems to be common in most acute and chronic neurodegenerative diseases associated with inflammation (reviewed in Domercq *et al*, 2013). IRF5 drives *de novo* expression of P2X4R by directly binding to the promoter region (Masuda *et al*, 2014). Here, we showed that *Irf5*, *Irf8*, and *P2x4r* mRNA expression is increased and is correlated at the peak as well as in the recovery phase of the EAE. Recent genome-wide SNP analysis has identified IRF8 as a susceptibility factor for multiple sclerosis (De Jager *et al*, 2009). In addition, genetic polymorphisms in human IRF5 that lead to the expression of various unique isoforms or higher expression of *Irf5* mRNA have been linked to autoimmune diseases, including MS (Kristjansdottir *et al*, 2008). IRF5 and IRF8 play a key role in the induction of pro-inflammatory cytokines,

Figure 7.

contributing to the plasticity and polarization of macrophages to a pro-inflammatory phenotype and initiation of a potent T(H)1-T(H)17 response that boost EAE disease progression (Krausgruber et al,

2011; Yoshida et al, 2014). In accordance, in vitro polarization of microglia toward a pro-inflammatory phenotype, not to an anti-inflammatory phenotype, upregulated P2X4R expression and

**Figure 7. P2X4R modulates myelin phagocytosis.**

A   Spinal cord sections of control mice, EAE mice, and EAE mice treated with TNP-ATP at the recovery phase stained for MBP (red) and Iba1 (green). Scale bar = 50 μm. Representative images from $n$ = 6 mice/group from two independent experiments.

B   Alexa 488-labeled myelin endocytosis (1 h, 37°C) in Iba1$^+$ microglia polarized in the absence or presence of TNP-ATP (10 μM). Scale bar = 50 μm.

C   Degradation of Alexa 488-labeled myelin by control, pro-inflammatory, and anti-inflammatory microglia at chase time 0, 3, and 6 days. *Left*, representative images at 0 and 6 days. Scale bar = 50 μm. *Right*, histogram shows the percentage of 488-myelin retained in the cells after the 3- and 6-day chase periods with respect to the 488-myelin at 0-h chase time.

D   Histogram represents the effect of TNP-ATP (10 μM), applied during differentiation and chase time (6 days), on myelin degradation. Myelin degradation at 6 days was expressed as fold change versus control microglia at the same chase time.

Data information: Data are presented as mean ± s.e.m. and were analyzed by Student's *t*-test. */#$P$ < 0.05, ##$P$ < 0.01. Symbols indicate significance versus control (*) or versus pro-/anti-inflammatory microglia (#). *In vitro* data come from $n$ = 4 independent experiments performed in duplicate.

function (Appendix Fig S3). However, the risk factor for MS of IRF5 and IRF8 contrasts with the protective role described here of P2X4R. Thus, although this receptor is activated during pro-inflammatory polarization, it is conceivable that P2X4R overexpression may help to resolve or counterbalance the inflammatory reaction by priming a subsequent anti-inflammatory response. Indeed, the presence of pro-inflammatory macrophages is a prerequisite for the successive emergence of anti-inflammatory macrophages and tissue homeostasis during wound healing and *Listeria monocytogenes* infections (Chazaud, 2014; Bleriot *et al*, 2015).

Antigen-presenting cells, including CNS microglia and perivascular macrophages, play pivotal roles in initiating Th17-cell development and transmigration through the BBB leading to EAE (Bartholomäus *et al*, 2009; Goldmann *et al*, 2013; Xiao *et al*, 2013; Yoshida *et al*, 2014). However, our results did not substantiate a direct (T-cell-mediated) or indirect (APC-dependent) role of P2X4R for the development of T-cell response and recruitment to the CNS, excluding any role of P2X4R on the onset of EAE disease. A role of P2X4R in recovery is also supported by the beneficial/detrimental effect of P2X4R manipulation in remyelination in LPC-treated slices, a model lacking adaptive immune activation.

The role of inflammation in promoting neural repair is gaining increasing recognition. Products of macrophages as well as of microglia, their CNS counterparts, facilitate the regeneration of axons (David *et al*, 1990; Yin *et al*, 2006) and promote remyelination in animal models of demyelination as their deficiency retards the process of remyelination (Kotter *et al*, 2005; Kondo *et al*, 2011; Miron *et al*, 2013; Sun *et al*, 2017; Cantuti-Castelvetri *et al*, 2018). However, the innate immune system capacity to restore myelination in the context of MS depends on microglia/macrophage polarization state. Thus, pro-inflammatory microglia/macrophage deactivation suppresses EAE acute phase (Starossom *et al*, 2012), whereas microglia/macrophage polarization to an anti-inflammatory phenotype is essential for efficient remyelination later on (Butovsky *et al*, 2006; Miron *et al*, 2013; Sun *et al*, 2017). Thus, a switch from a pro-inflammatory to an anti-inflammatory dominant polarization of microglia/macrophage is critical in the repair process, and therefore, manipulating polarization phenotypes of microglia/macrophage might be a promising therapeutic strategy for treating MS. We here demonstrate that blocking P2X4R exacerbated a switch to a pro-inflammatory phenotype and increased neurological deterioration in the recovery phase, whereas its potentiation with IVM increased anti-inflammatory polarization and ameliorated clinical signs. Resident microglia and monocytes contribute differentially to EAE induction (Ajami *et al*, 2011;

Yamasaki *et al*, 2014), whereas few studies have addressed their specific contribution to remyelination (Lampron *et al*, 2015). The experiments described in this paper do not allow us to discriminate between microglia and monocyte-derived macrophages, and further experiments are necessary to define the role played by P2X4R in the two cell populations.

The benefits of microglia/macrophage may be attributed to being required in clearing myelin debris after a demyelinating episode (Kotter *et al*, 2006; Neumann *et al*, 2009; Lampron *et al*, 2015; Cantuti-Castelvetri *et al*, 2018), as well as their release of a variety of growth factors into the injured CNS that favor oligodendrocyte differentiation (Miron *et al*, 2013). Phagocytosis of myelin is more robust in anti-inflammatory microglia than in pro-inflammatory microglia (Durafourt *et al*, 2012; Healy *et al*, 2016). We also detected an increase in myelin endocytosis as well as in the subsequent myelin degradation in anti-inflammatory microglia, and a decrease in pro-inflammatory microglia. Moreover, we demonstrated here that P2X4R blockade or potentiation modulates the effect of polarization on phagocytosis. However, the opposite interpretation is also possible. Thus, phagocytosis of myelin controls microglia/macrophage inflammatory response (Kroner *et al*, 2014). Recently, it has been described that phagocytosis of myelin in aged microglia/macrophages after demyelination results in cholesterol accumulation in these cells, leading to a maladaptive inflammatory response with inflammasome activation that impairs remyelination (Cantuti-Castelvetri *et al*, 2018).

Our data showed that IVM also potentiates myelin engulfment and degradation in control microglia. Previous studies have described that P2X4R-mediated endolysosomal Ca$^{2+}$ release is involved in vacuolation and endolysosomal membrane fusion with lysosomes (Cao *et al*, 2015) which could control phagocytosis. In accordance, we observed that P2X4R induces endosome–lysosome fusion and lysosome pH acidification, a pivotal step for enzymatic degradation of material delivered by phagocytic pathways. Thus, it is possible that these strategically located P2X4Rs could directly modulate myelin phagocytosis. Whether IVM potentiation of phagocytosis is the mechanism controlling microglia polarization or the opposite requires further studies.

On the other hand, previous data on literature demonstrated that OPC differentiation and myelination in the CNS are controlled by highly regulated sequences of molecular interactions with neurotransmitters released by axons, growth factors, neuregulins, integrins, and cell adhesion molecules. Among all, it is well known that BDNF enhances oligodendrocyte differentiation and myelination (Wong *et al*, 2013). A source of BDNF promoting oligodendrogenesis after white matter ischemic insults is astrocytes (Miyamoto

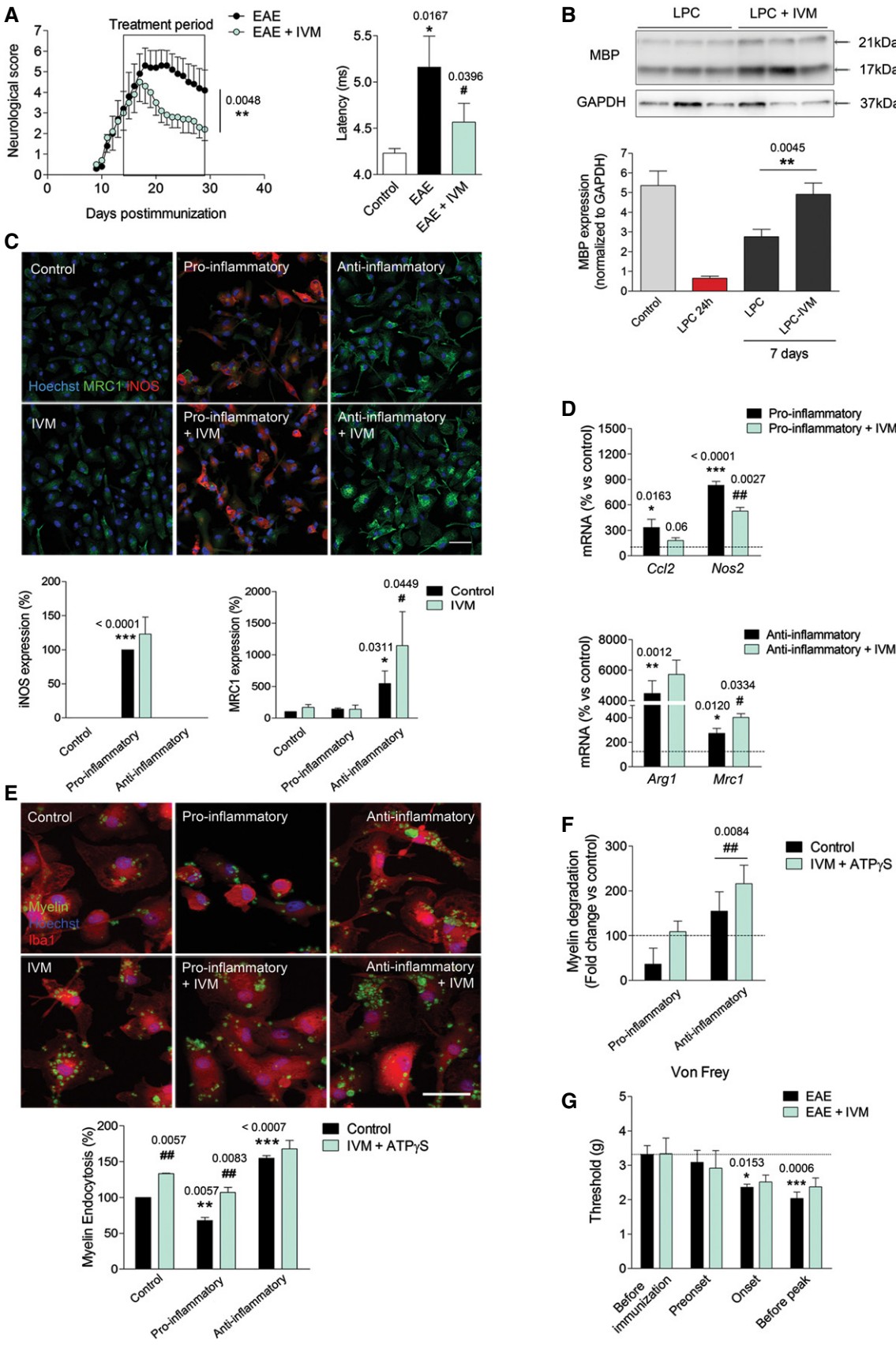

Figure 8.

◀

**Figure 8.  Potentiation of P2X4R favors remyelination and ameliorates EAE.**

A  *Left*, neurological score of EAE (*n* = 5) and ivermectin (IVM)-treated EAE mice (1 mg/kg) (*n* = 5 mice/group; one representative experiment of two independent experiments). *Right*, axon conduction latency in the corticospinal tract of control and vehicle- and IVM-treated EAE mice. Symbols indicate significance versus control (*) or versus EAE (#).

B  Effect of IVM (3 μM)- on lysolecithin (LPC)-induced demyelination in cerebellum organotypic slices, as analyzed by MBP densitometry (*n* = 5).

C  Staining for iNOS (red) and MRC1 (green) in control and differently activated microglia in the absence or presence of IVM (3 μM) (*n* = 3). Scale bar = 50 μm. Symbols indicate significance versus control (*) or versus anti-inflammatory microglia (#).

D  qPCR quantification of pro-inflammatory genes (*Ccl2* and *Nos2*) and anti-inflammatory genes (*Arg1* and *Mrc1)* in different activated microglia in the absence or presence of IVM (3 μM) (*n* = 3). Symbols indicate significance versus control (*) or versus pro-/anti-inflammatory microglia (#).

E  Control, pro-inflammatory, and anti-inflammatory microglia endocytosis (1 h) of Alexa 488-labeled myelin in the absence or presence of IVM (3 μM) plus ATPγS (10 μM) (*n* = 4). Scale bar = 40 μm. Symbols indicate significance versus control (*) or between IVM + ATPγS-treated and non-treated cells (#).

F  Myelin degradation at 6 days, expressed as fold change versus control microglia at the same chase time, in the absence and in the presence of IVM (3 μM) + ATPγS (10 μM) (*n* = 4).

G  Withdrawal threshold (grams) of mechanical stimulation by von Frey filaments applied to the mouse hind paw in non-treated (*n* = 6) and IVM-treated mice (*n* = 7) at specific stages of EAE progression.

Data information: Data are presented as mean ± s.e.m. Statistics were performed with Mann–Whitney *U*-test (neurological score), one-way ANOVA (A, D, E, F), and Student's *t*-test (B, C). *$^{/\#}P$* < 0.05, **$^{/\#\#}P$* < 0.01, ***$P$* < 0.001.

Source data are available online for this figure.

---

*et al*, 2015). However, microglia are also another important source of BDNF in physiological conditions and after injury (Dougherty *et al*, 2000; Parkhurst *et al*, 2013), and microglia P2X4R activation has been linked to BDNF release, causing tactile allodynia (Ferrini *et al*, 2013). We showed here that BDNF production by microglia was increased in anti-inflammatory microglia, an effect significantly reduced by TNP-ATP treatment. In addition, *Mbp* levels after EAE strongly correlated with *Bdnf* levels and were dramatically reduced in the recovery phase of EAE after TNP-ATP treatment. These data are only correlative, so we not exclude the role of other factors secreted by microglia after P2X4R activation to EAE remyelination.

Manipulating innate immune system to promote repair might be a promising therapeutic strategy for treating MS. The results of our study identify P2X4R as a key modulator of microglia/macrophage polarization and support the use of IVM to potentiate a microglia/macrophage switch that favors remyelination in MS. It is important to note that anti-helminthic host responses are based on anti-inflammatory macrophage polarization (Satoh *et al*, 2010), and thus, it is conceivable that the mechanism described here could be added to the IVM therapeutic effects against helminths. The fact that IVM is already used as an anti-parasitic agent in humans will facilitate challenging this drug in clinical trials in that demyelinating disease.

# Materials and Methods

## Animals

All experiments were performed according to the procedures approved by the Ethics Committee of the University of the Basque Country (UPV/EHU). Animals were handled in accordance with the European Communities Council Directive. Mice were kept under conventional housing conditions (22 ± 2°C, 55 ± 10% humidity, and 12-h day/night cycle) at the University of the Basque Country animal facilities. All possible efforts were made to minimize animal suffering and the number of animals used. Generation of P2X4$^{-/-}$ mice was described previously (Sim *et al*, 2006).

## Cell cultures

### Microglia and OPC culture

Primary mixed glial cultures were prepared from the cerebral cortex of neonatal rats and mice (P0–P2) as previously described (Domercq *et al*, 2007). After 10–15 days in culture, microglia were isolated by mechanical shaking (400 rpm, 1 h) as previously described (Domercq *et al*, 2007). The remaining oligodendrocyte progenitor cells (OPCs) present on the top of the confluent monolayer of astrocytes in the flasks were dislodged by shaking flasks overnight at 400 rpm. The final cell suspension was collected, centrifuged, and resuspended in a chemically defined high-glucose Dulbecco's modified Eagle's medium supplemented with 100 μg/ml transferrin, 60 ng/ml progesterone, 40 ng/ml sodium selenite, 5 μg/ml insulin, 16 μg/ml putrescine, and 100 μg/ml BSA.

Microglial cells were polarized according to previous protocols (Durafourt *et al*, 2012) with minor modifications. To generate pro-inflammatory microglia, cells were treated with GM-CSF (5 ng/ml; Peprotech) for 5 days followed by 24-h treatment with LPS (10 ng/ml) and IFNγ (20 ng/ml; Peprotech). To generate anti-inflammatory microglia, cells were treated with M-CSF (20 ng/ml; Peprotech) for 5 days followed by 24-h treatment with IL-4 (20 ng/ml; Peprotech) and IL-13 (50 ng/ml; Peprotech). Microglia-conditioned medium was collected and centrifuged (270 *g*, 5 min). OPCs were treated with microglia-conditioned medium for 3 days at 37°C. Polarizing factors alone were directly applied to OPCs as a control. Calcium measurements were performed at 37°C using Fluo-4 calcium indicator in a Leica LCS SP2 AOBS confocal microscope.

### T cells

Mouse splenocytes were obtained by mashing the disintegrated organs through cell strainers into PBS. Cell suspension was freed from erythrocytes by incubation with ACK lysing buffer (155 mM NH$_4$Cl, 10 mM KHCO$_3$, 100 μM EDTA, pH ~7.2). Cytokine measurement and cell proliferation assay are described in Appendix.

### Cerebellar organotypic cultures

Cultures were prepared from cerebellar sections of P5–P7 Sprague Dawley rat pups according to previously described procedures (Cavaliere *et al*, 2010). Slices (350 μm) were maintained in medium

consisting in 50% basal medium with Earle's salt, 25% HBSS, 25% inactivated horse serum, 5 mg/ml glucose (Panreac), 0.25 mM L-glutamine (Sigma-Aldrich) at 37°C in an atmosphere of humidified 5% $CO_2$. Demyelination was induced at 14 days *in vitro* by lysolecithin (0.5 mg/ml; 17 h). Slices were allowed to remyelinate during 1 week, and remyelination was analyzed by Western blot using antibodies to MBP (#SMI-99P; Covance).

### EAE induction

EAE was induced in 8- to 10-week-old female P2X4$^{-/-}$ (backcrossed for > 15 generations onto C57Bl/6 background; Sim *et al*, 2006) and their wild-type littermate C57Bl/6 mice by subcutaneous immunization with 300 μl of myelin oligodendrocyte glycoprotein 35–55 (MOG; 200 μg; Sigma) in incomplete Freund's adjuvant supplemented with 8 mg/ml *Mycobacterium tuberculosis* H37Ra. Pertussis toxin (500 ng; Sigma) was injected on the day of immunization and again 2 days later. Motor function was recorded daily and scored from 0 to 8 following standard scales (Matute *et al*, 2007). Mice were daily treated with TNP-ATP (10 mg/kg, i.p.; Tocris), ivermectin (1 mg/kg, i.p.; Sigma), or vehicle from EAE onset to the end of the experiment except for the experiments designed to check the effect of TNP-ATP on immune priming, where TNP-ATP was administered daily from day 0 to EAE peak. All mice were randomized before the immunization and before the appearance of EAE symptoms. Conduction velocity of the corticospinal tract was assessed at the end of the experiment in anesthetized mice with tribromoethanol (240 mg/kg, i.p.; Sigma) using stimulatory and recording electrodes placed in the primary motor cortex and in the vertebral canal at the L2 level, respectively (Matute *et al*, 2007). Neurological score and latency recording were undertaken by readers blinded to the study. Disease phases were assigned according to days after onset as follows: peak, 6–10 days after onset; and recovery, score stabilized and 18–30 days after onset.

### Pain assessment

Mechanical allodynia was assessed by an e-VF Electronic von Frey aesthesiometer (Ugo Basile SRL) at different stages of EAE: before immunization, EAE preonset (5 dpi), onset (10–11 dpi), and before peak (15 dpi). Mice, placed upon an elevated wire mesh surrounded by a Perspex box, were exposed to increasing mechanical pressure to the plantar hind paw through a metal filament. Withdrawal threshold was measured automatically from the initiation of mechanical stimulus to withdrawal of the paw three times in both left and right hind paws separated by at least 10 min between each stimulus. Mean results for each animal were calculated.

### Immunochemistry

Cells were fixed in 4% PFA in PBS for 20 min and processed for conventional immunocytochemistry. For tissue, adult mice were deeply anesthetized with chloral hydrate (500 mg/kg, i.p.) and transcardially perfused with 0.1 M sodium phosphate buffer, pH 7.4, followed by 4% PFA in the same buffer. Antibodies used are described in Appendix.

Images were acquired using a Leica TCS STED SP8 confocal microscope or a Zeiss AxioVision microscope with the same settings

for all samples within one experimental group. Olig2$^+$ cells in corpus callosum and in longitudinal sections of spinal cord and Iba1$^+$ cells in longitudinal sections of spinal cord were counted blindly using a 40× objective in an AxioVision microscopy (Zeiss). At least four different fields from three slices per animal were counted from each mouse. To quantify microglia polarization, immunoreactivity of iNOS and MRC1 was calculated with the ImageJ software (NIH) and normalized to the number of cells (eight fields per coverslip from at least four different experiments performed in triplicate). To analyze the effect on OPC differentiation, MBP$^+$ cells were counted and the results were expressed in percentage versus total cells (15 fields per coverslip were analyzed by two observers from n = 3 different experiments performed in triplicate). To quantify microglia and oligodendrocyte expression of P2X receptors, regions of interest (ROIs) were generated with ImageJ software (NIH) in IB4$^+$ and Olig2$^+$ cells (12–15 cells per culture from three independent experiments).

### Western blot

Total protein was extracted from microglia by scraping the cells in SDS/sample buffer. Tissue from cerebellar organotypic slices was directly heated at 100°C in sample buffer (7 min). Samples were loaded and size-separated by electrophoresis using Criterion TGX Precast 12% gels and transferred to Trans-Blot Turbo Midi PVDF Transfer Packs (Bio-Rad, Hercules, USA). Membranes were blocked in 5% skimmed milk and 5% serum in Tris-buffered saline/0.05% Tween-20 (TBS-T) and proteins detected by specific primary antibodies to BDNF (#sc-547, 1:200; Santa Cruz), to MBP (#SMI-99P, 1:2,000, Covance), to GAPDH (#MAB374, 1:2,000; Millipore), and to β-actin (#A2066, 1:1,000; Sigma), followed by secondary peroxidase-coupled goat anti-rabbit antibodies (#A6154, 1:2,000; Sigma) or sheep anti-mouse antibodies (#A6782, 1:2,000; Sigma). After washing, blots were developed using an enhanced chemiluminescence detection kit according to the manufacturer's instructions (SuperSignal West Dura or Femto, Pierce). Images were acquired with a ChemiDoc MP system (Bio-Rad) and quantified using ImageJ software. Values of BDNF and MBP were normalized to corresponding β-actin and GAPDH signal, respectively.

### Myelin phagocytosis and lysosomal pH measurement

Rat and mouse myelin was isolated as previously described (Norton & Poduslo, 1973). Briefly, brain was mechanically homogenized in 0.32 M sucrose and subjected to repeated sucrose gradient centrifugation and osmotic shocks to separate myelin from other cellular components. Myelin was incubated with Alexa 488-NHS dye (A2000 Life Technologies) for 1 h 45 min at RT in PBS (pH 8). Dyed myelin was dialyzed for removing dye excess, resuspended in PBS (pH 7.4), vortexed for 60 s for fragmentation in homogeneous size aggregates, and added to microglia culture medium (1:100 dilution).

To evaluate myelin endocytosis, microglia were incubated with Alexa 488-NHS-labeled myelin for 1 h at 37°C, rinsed, and fixed. To evaluate myelin degradation, cells were subsequently chased at 3 and 6 days for rat microglia and at 3 days for mouse microglia. Cells were fixed with 4% PFA and stained using antibodies to Iba1 (#019-19741, 1:500; Wako) and Hoechst 33258. Myelin was quantified on Iba1$^+$ cells using ImageJ on individual microglial cells outlined with the Iba1 immunostaining as the defining parameter for the ROIs (at

least 50 cells were analyzed in each experiment). Identical acquisition parameters were used for image capture of individual experiments.

The measurement of lysosomal/endosomal pH by confocal microscopy is based on the use of the ratio of the pH-sensitive fluorescein fluorescence to pH-insensitive rhodamine fluorescence as previously described (Majumdar et al, 2007). The same dye was used to measure lysosome area. Briefly, cells were incubated for 16 h with 5 mg/ml dextran conjugated to both fluorescein and rhodamine (70,000 mol. wt.; ThermoFisher), washed thoroughly, and then further incubated for 2 h to chase dextran into lysosomes. Cells were then examined by confocal imaging at 37°C. Lysosome fluorescence and lysosome area were quantified in defined ROIs corresponding to individual lysosomes using Fiji software. For pH measurements, the ratio of fluorescein to rhodamine fluorescence was determined. For all experimental sets, cross-talk of the fluorophores was negligible. Calibration curves were generated after fixing and equilibrating the fluorescein–rhodamine–dextran-loaded cells to a range of buffer pH values. We quantified about 40 ROIs per field (63× photographs) from at least 10 fields per condition in two independent experiments performed in duplicate.

### Electrophysiology

Cortical slices (300 μm thick) were prepared from the brain of P15–P20 PLP-DsRed and CXCR3-GFP mice to record oligodendrocytes and microglia, respectively. Slices were obtained in ice-cold solution containing (in mM): 215 sucrose, 2.5 KCl, 26 $NaHCO_3$, 1.6 $NaH_2PO_4$, 1 $CaCl_2$, 4 $MgCl_2$, 4 $MgSO_4$, 20 glucose, and 1.3 ascorbic acid bubbled with 95% $O_2$/5% $CO_2$, pH 7.4. Slices were then stored for at least 1 h at 32°C in artificial cerebrospinal fluid (aCSF) that contained (in mM) 124 NaCl, 2.5 KCl, 25 $NaHCO_3$, 1.2 $NaH_2PO_4$, 1.25 $CaCl_2$, 2.6 $MgCl_2$, and 10 glucose. Standard whole-cell recordings of microglia and oligodendrocytes (Vh = −70 mV) were performed at 37°C on a Leica microscope (Leica DM LFSA) using the MultiClamp 700B amplifier (Axon). Recordings were low-pass-filtered at 2 kHz, digitized at 5 kHz, and stored as data files on a computer using the pClamp 8.2 program (Axon Instruments, CA) for later analysis. The extracellular bath solution contained (in mM) 124 NaCl, 26 $NaHCO_3$, 1 $NaH_2PO_4$, 2.5 KCl, 2 $MgCl_2$, 2.5 $CaCl_2$, and 10 glucose, bubbled with 95% O2/5% $CO_2$, pH 7.4. ATP was applied through a 5242 microinjector (Eppendorf) in divalent cation-free extracellular solutions to maximize purinergic currents. Patch clamp pipettes (3–5 MΩ) were filled with a solution containing (in mM) 135 KCl, 4 NaCl, 0.7 $CaCl_2$, 4 MgATP, 10 HEPES, 10 BAPTA, and 0.5 $Na_2$-GTP, pH 7.3.

### FACS

For flow cytometric analyses, cells were stained in FACS buffer containing 0.1% BSA and 1 mM EDTA with fluorochrome-conjugated monoclonal antibodies. Details about antibodies used are described in Appendix. Stained cells were washed and resuspended in 300 μl FACS buffer. Cells were measured using a BD LSRFortessa, and data were analyzed with FlowJo software (Tree Star).

For microglia sorting, samples were sorted using CD11b (#101205, 1:100; Biolegend), CD45 (#103134, 1:100; Biolegend), Ly6C (#128012, 1:100; Biolegend), Ly6G (#127622, 1:100; Biolegend), and CCR2 (#FAB5538, 1:100; R&D Systems) to distinguish

between resident microglia (CD11b⁺/CD45^low) and invading macrophages (CD11b⁺/CD45^high; Szulzewsky et al, 2015). Cells were sorted using a FACSAria IIIu (BD Bioscience).

### qPCR and gene expression profiling

Total RNA from control and EAE lumbar spinal cords and from microglia cultures was isolated using TRIzol (Invitrogen) according to the manufacturer's instructions. RNA from sorted microglia was isolated using RNeasy Plus Micro Kit (Qiagen). Subsequently, cDNA synthesis was conducted using SuperScript III retrotranscriptase (200 U/μl; Invitrogen) or AffinityScript Multiple Temperature cDNA Synthesis Kit (Agilent Technologies Inc.) and random hexamers as primers (Promega). cDNA from EAE experiments was processed via Fluidigm real-time PCR analysis and GenEx software, and the results were depicted as relative gene expression according to the $\Delta\Delta C_q$ method ($2^{-\Delta\Delta C_t}$) and expressed in base 2 logarithmic scale. For in vitro experiments, real-time quantitative PCRs were performed with SYBR Green using a Bio-Rad CFX96 Real-Time PCR Detection System as described previously (Domercq et al, 2016).

### PET imaging

[¹⁸F]BR-351, the matrix metalloproteinase inhibitor used for PET imaging (¹⁸F-MMPi), was prepared from its tosylate precursor according to a previously reported procedure (Wagner et al, 2011) using a TRACERlab FX_FN synthesis module (GE Healthcare). Radiochemical yields (non-decay-corrected) were in the range 12–16%, and radiochemical purity was always > 95% at the time of injection. Naïve control mice and EAE mice ± TNP-ATP (treatment from day 0 to day 17 postimmunization) at the peak of the EAE were subjected to μPET imaging using the radiotracer [¹⁸F]BR-351 for assessment of MMP activity. Thirty-minute static μPET scans were acquired for 100 min after intravenous injection of 10 MBq. CT acquisitions were performed to provide anatomical information and the attenuation map for reconstruction using 2DOSEM. PET images were co-registered to the anatomical data of the CT of the same mouse, and volumes of interest (VOIs) were manually drawn in the inner part of the brain and spinal cord. The PET signal was expressed as percentage of injected dose per gram of tissue (%ID/g).

### Statistical analysis

Data are presented as mean ± s.e.m. with sample size and number of repeats indicated in the figure legends. Comparisons between two groups were analyzed using paired Student's two-tailed t-test for data coming from in vitro experiments and unpaired Student's two-tailed t-test for data coming from in vivo experiments, except in MOG-EAE experiments where statistical significance in neurological score was determined by Mann–Whitney U-test. Comparisons among multiple groups were analyzed by one-way analysis of variance (ANOVA) followed by Bonferroni's multiple comparison tests for post hoc analysis. Statistical significance was considered at $P < 0.05$.

### Study approval

Organotypic cultures, primary cultures, electrophysiology protocols, and EAE experiments in wild-type or P2X4⁻/⁻ mice were approved

## The paper explained

### Problem

Promoting endogenous remyelination represents an attractive target for regeneration in multiple sclerosis (MS). A major component of remyelination in the CNS is a robust innate immune response. Although microglia and macrophages are implicated in CNS autoimmune disease via secretion of toxic molecules and antigen presentation to cytotoxic lymphocytes, they also exhibit regenerative properties through the phagocytosis of myelin debris and secretion of growth and neurotrophic factors that promote oligodendrocyte progenitors differentiation and myelination. Thus, identifying molecular targets that could switch microglia/macrophage activation/polarization could be therapeutically relevant. We have here analyzed the role of P2X4 receptor, as we detected previously P2X4R+ reactive microglia/macrophages to be present in EAE as well as in MS demyelinating lesions.

### Results

To study the role played by P2X4R in MS pathophysiology, we tested the effects of pharmacological or genetic blocking of P2X4 as well as the effect of the allosteric modulator ivermectin, an enhancer of P2X4R function, in the experimental autoimmune encephalitis mice (EAE), a chronic animal model of MS. Treating mice with TNP-ATP, an antagonist of P2X4R, or knockout for the P2X4 gene showed exacerbated EAE neurological symptoms, whereas administration of ivermectin, an enhancer of P2X4R function, ameliorated neurological damage. Detailed FACS immune analysis in combination with PET imaging demonstrated that P2X4R modulation did not interfere with the efficacy of immunization and the cellular adaptive immune response against MOG. Rather, the beneficial effect of ivermectin, administered after the onset of the disease, was clear in the recovery phase and was associated with a switch in microglia polarization that favors myelin phagocytosis and remyelination. This conclusion is also supported by the results obtained in another model of MS, which lack the adaptive immune activation, the lysolecithin-induced demyelination. Thus, we propose that ivermectin could have potential benefit to promote remyelination in MS lesions.

### Impact

Available therapies for MS include immunomodulators that reduce relapses, but they do not halt the neuronal damage and the remyelination. Understanding chronic progression of MS, more than autoimmune-driven relapses, and understanding the disease pathways that allow recovery in the animal models is certainly relevant to design alternative interventions to treat specifically primary and secondary progressive MS.

by the Comité de Ética y Bienestar Animal (Animals Ethics and Welfare Committee) of the University of the Basque Country. All the experiments were conducted in accordance with the Directives of the European Union on animal ethics and welfare.

Expanded View for this article is available online.

## Acknowledgements

This work was supported by Merck Serono (a business of Merck KGaA, Darmstadt, Germany) grants (to M.D. and C.M. and to T. M.); Spanish Ministry of Education and Science (SAF2013-45084-R and SAF2016-75292-R); Basque Government (fellowship to N.V-V.); University of the Basque Country (UPV/EHU; fellowship to A.Z. and J.G.); and Centro de Investigación Biomédica en Red, Enfermedades Neurodegenerativas (CIBERNED). The authors would like to thank to S. Marcos and L. Colás for technical assistance in culture and in PET imaging, respectively.

## Author contributions

AZ conducted the microglia polarization experiments and the EAE experiments, and analyzed and interpreted the data; NV-V performed the first EAE experiments; AZ, BR and ML performed the FACS study and the *in vitro* analysis on T cells; JG performed the myelin phagocytosis study; AP performed the remyelination studies on organotypic slices; AM, KRP, and JL performed the synthesis of the radioligand [18F]BR-351 and the PET imaging study; AP-S and MD performed the electrophysiology recordings in slices; EC-Z, TM, and FK-N contributed to designing phagocytosis studies and immune cell analysis; FR developed the P2X4−/− mice; CM designed the study, conducted and supervised EAE experiments, and wrote the manuscript; and MD designed the study, supervised all experiments, analyzed and interpreted the data, and wrote the manuscript.

## Conflict of interest

The authors declare that they have no conflict of interest.

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
