## [Review Process File · EMBO Molecular Medicine]

P2X4 receptor controls microglia activation and favours remyelination in autoimmune encephalitis

Alazne Zabala, Nuria Vazquez-Villoldo, Björn Rissiek, Jon Gejo, Abraham Martin, Aitor Palomino, Alberto Perez-Samartín, Krishna R Pulagam, Marco Lukowiak, Estibaliz Capetillo-Zarate, Jordi Llop, Tim Magnus, Friedrich Koch-Nolte, Francois Rassendren, Carlos Matute and María Domercq

Review timeline:	Submission date:	04 December 2017
	Editorial Decision:	19 January 2018
	Revision received:	04 May 2018
	Editorial Decision:	25 May 2018
	Revision received:	05 June 2018
	Accepted:	08 June 2018

Editor: Céline Carret

Transaction Report:

1st Editorial Decision

19 January 2018

Thank you for the submission of your manuscript to EMBO Molecular Medicine. We have now heard back from the two referees whom we asked to evaluate your manuscript.

You will see that while they appreciated the study and translational relevance, they also would like to see more mechanism of action, better controls and further detailed explanations. I would like to encourage you to address these as requested.

We would welcome the submission of a revised version within three months for further consideration and would like to encourage you to address all the criticisms raised as suggested to improve conclusiveness and clarity. Please note that EMBO Molecular Medicine strongly supports a single round of revision and that, as acceptance or rejection of the manuscript will depend on another round of review, your responses should be as complete as possible.

I look forward to receiving your revised manuscript.

***** Reviewer's comments *****

Referee #1 (Comments on Novelty/Model System for Author):

The authors use a wide range of appropriate and sophisticated techniques to address the question of the role of microglial P2X4R in demyelination.

Referee #1 (Remarks for Author):

The paper significantly improves understanding of the mechanisms of microglia responses during demyelination. Specifically, the study demonstrates a key role for purinergic receptor P2X4 (P2X4R) in microglia/macrophages during autoimmune inflammation. Importantly, the study shows that genetic or pharmacological blockade of P2X4R signaling exacerbates disease in EAE, the mouse model of MS. Furthermore, P2X4R regulated remyelination and repair. These important findings support the possibility that P2X4R may be a potential therapeutic target in MS

Some minor comments:

1) Levels of P2x4R expression were increased at the peak of the disease and remained elevated during the recovery phase of EAE (Fig. 1), which suggest against either a specific protective or destructive role for P2X4R, since they are elevated during both damage and repair? How does this square with the subsequent findings on the regulatory role of P2X4R?

For example, the gene expression (Fig. 4) indicates that blockade of P2X4R with TNPATP did not significantly alter anti-inflammatory gene expression, which is associated with repair, but did significantly increase pro-inflammatory gene expression, which is associated with the peak of EAE.

2) TNP-ATP was used to antagonise P2X4R, but TNP-ATP is a non-selective P2X antagonist. Although they also used the P2X4R KO to show the effects are mediated primarily through P2X4R, it would still have been good to use some other more specific P2X4R antagonists. Also, what was the concentration in the brain?

3) Also with TNP-ATP, the data in figure 2 indicate that blockade of P2X4R decreases the neurological score, but the gene expression (Fig. 4) indicates that blockade of P2X4R with TNP-ATP significantly increase pro-inflammatory gene expression, which is associated with the peak of EAE and considered detrimental. Can this be explained?

Referee #2 (Comments on Novelty/Model System for Author):

This is a thorough study that might have important practical applications, especially because one of the drugs investigated, ivermectin, is already used or the treatment of human diseases. Clarity might be improved.

Referee #2 (Remarks for Author):

This MS highlights the role of the P2X4R as an anti-inflammatory receptor that helps to damp down inflammation in a model of experimental autoimmune encephalitis (EAE) and to promote remyelination. Processes involved in the pathogenesis of EAE are a matter of hot debate in view of their obvious relevance for multiple sclerosis, but unfortunately little progress has been made towards elucidation of molecular pathways involved. In this context, the P2X4R is attracting interest as one of the P2Rs whose expression is increased at sites of neurological damage in EAE. However, role of this receptor in EAE is unknown. The study by Zabala et al. aims at clarifying this issue. The main finding is that the P2X4R turns out to have an anti-inflammatory, protective, role in this disease. The study is thorough, informative and of potential therapeutic relevance. My main criticism relates to the lack of clear mechanistic explanation for the role of P2X4. The Authors show that P2X4 blockade with TNP-ATP reduces myelin endocytosis, pointing to phagocytosis modulation as a possible mechanism, yet this was not investigated in depth. For example, I wonder why no phagocytosis experiments were performed with microglia from P2X4-KO mice. Also, the finding that IVM treatment does not increase myelin endocytosis in anti-inflammatory microglia is

disturbing. In this regard, IVM should also be tested in the P2X4-KO mice: here no effect is anticipated. In addition, the Authors briefly addressed the potential role of BDNF. This is an interesting point since BDNF release is linked to P2X4 activation. However, this was not investigated in depth, and even basic controls in the P2X4-KO model were not performed. Additional minor points for the Authors to consider are detailed below.

1. Please, for the sake of the lay reader, explain what MOG35-55 is, and the rationale for its use, in the Result section.
2. Please, add "+" superscript to CD4 and CD8 in several places throughout the MS.
3. At pg 9, line 19, "is" should be "are".
4. It is kind of unusual that Dr Rassendren is thanked for the gift of the P2X4-KO mice when he is in fact a co-author.
- 5) A pg 24 (Pain assessment): I do not understand what the reference to "Ugo Basile" means.
- 6) Sentence at pg 29, lines 9-12, starting with "Naive..." should be amended.
- 7) Images in Fig. 2B are not really convincing. I urge the Authors to provide better images.

1st Revision - authors' response

04 May 2018

RESPONSE TO REVIEWERS

We appreciate all the points raised by the reviewers which helped us to improve our study. We have carefully addressed them as explained in detail below. In particular, we have performed new experiments to elucidate the mechanism by which P2X4R modulate microglia reaction. In addition to controlling BDNF release as previously described (Fig EV2), we have analyzed whether P2X4R could directly control lysosome function and thus modulate directly myelin phagocytosis. We indeed have observed that P2X4R potentiation with ivermectin induces lysosome acidification (Fig. EV6). Further analysis, out of the scope of the present study, are necessary to determine the mechanism by which P2X4R controls lysosome function.

Point by point response to reviewers

Referee #1

Some minor comments:

1) Levels of P2x4R expression were increased at the peak of the disease and remained elevated during the recovery phase of EAE (Fig. 1), which suggest against either a specific protective or destructive role for P2X4R, since they are elevated during both damage and repair? How does this square with the subsequent findings on the regulatory role of P2X4R?

For example, the gene expression (Fig. 4) indicates that blockade of P2X4R with TNP-ATP did not significantly alter anti-inflammatory gene expression, which is associated with repair, but did significantly increase pro-inflammatory gene expression, which is associated with the peak of EAE.

Response:

Blockage of P2X4R with TNP-ATP induced a significant increase in pro-inflammatory gene expression only at the recovery phase. At the peak of the disease there was no significant changes in pro-inflammatory gene expression (see new Fig EV2). The lack of effect of P2X4R blockage on the inflammatory reaction at EAE peak is in accordance with the lack of effect on immune priming after prolong treatment (from day 0 after immunization, Fig. 3) and points to a crucial role of microglial P2X4 receptor mainly at the recovery phase. Innate immune response is essential to phagocyte myelin, a key process crucial to proceed with remyelination. Our data (see Fig 8 and new Fig EV6) supports the idea that P2X4R activation increases myelin endocytosis and degradation at lysosomes, probably by inducing lysosome acidification (Fig EV6).

2) TNP-ATP was used to antagonise P2X4R, but TNP-ATP is a non-selective P2X antagonist. Although they also used the P2X4R KO to show the effects are mediated primarily through P2X4R,

it would still have been good to use some other more specific P2X4R antagonists. Also, what was the concentration in the brain?

Response:

The specificity of the pharmacological tools has been demonstrated using the P2X4^{-/-} mice. Thus, drugs have no effect on P2X4^{-/-} mice (see Fig 2). Unfortunately, there is no a selective and potent antagonist of P2X4R with solubility in water. 5-BDBD works as a selective P2X4 receptor antagonist. However, the compound displays a very low water-solubility, which limits its application using systemic injection. An exception is the new compound NP-1815-PX (5-[3-(5-thioxo-4H-[1,2,4]oxadiazol-3-yl)phenyl]-1H-naphtho[1, 2-b][1,4]diazepine-2,4(3H,5H)-dione) which is a potent and selective antagonist of P2X4R (PMID:27576299). However, the compound is not commercial.

A previous study reported IVM brain accumulation after chronic IVM i.p. injection in mice (3 mg/kg; see Fig 2 in PMID:25004078). No previous study has analyzed the levels of TNP-ATP after i.p. injection and we did not have the methodological tools to analyze that. However, the severe CNS inflammation in EAE leads to BBB breakdown and thus increasing permeability to drugs.

3) Also with TNP-ATP, the data in figure 2 indicate that blockade of P2X4R decreases the neurological score, but the gene expression (Fig. 4) indicates that blockade of P2X4R with TNP-ATP significantly increase pro-inflammatory gene expression, which is associated with the peak of EAE and considered detrimental. Can this be explained?

Response:

Blockage of P2X4R with TNP-ATP exacerbates EAE.

Referee #2

This MS highlights the role of the P2X4R as an anti-inflammatory receptor that helps to damp down inflammation in a model of experimental autoimmune encephalitis (EAE) and to promote remyelination. Processes involved in the pathogenesis of EAE are a matter of hot debate in view of their obvious relevance for multiple sclerosis, but unfortunately little progress has been made towards elucidation of molecular pathways involved. In this context, the P2X4R is attracting interest as one of the P2Rs whose expression is increased at sites of neurological damage in EAE. However, role of this receptor in EAE is unknown. The study by Zabala et al. aims at clarifying this issue. The main finding is that the P2X4R turns out to have an anti-inflammatory, protective, role in this disease. The study is thorough, informative and of potential therapeutic relevance. My main criticism relates to the lack of clear mechanistic explanation for the role of P2X4. The Authors show that P2X4 blockade with TNP-ATP reduces myelin endocytosis, pointing to phagocytosis modulation as a possible mechanism, yet this was not investigated in depth. For example, I wonder why no phagocytosis experiments were performed with microglia from P2X4-KO mice. Also, the finding that IVM treatment does not increase myelin endocytosis in anti-inflammatory microglia is disturbing. In this regard, IVM should also be tested in the P2X4-KO mice: here no effect is anticipated. In addition, the Authors briefly addressed the potential role of BDNF. This is an interesting point since BDNF release is linked to P2X4 activation. However, this was not investigated in depth, and even basic controls in the P2X4-KO model were not performed. Additional minor points for the Authors to consider are detailed below.

Regarding BDNF, we have performed new control experiments to demonstrate that P2X4 receptor is linked to BDNF release. We demonstrated that P2X4R activation with IVM induces an increase in BDNF production in microglia from WT mice, an effect absent in P2X4^{-/-} microglia. This information has been added to the new Fig EV4. In addition, we have added new data (Fig EV4) showing that *Bdnf* mRNA is reduced in control and EAE P2X4^{-/-} mice vs wild type mice.

Regarding mechanism, two possibilities are discussed in the manuscript. One is that P2X4R modulate microglia polarization and indirectly, myelin phagocytosis. Indeed, P2X4R control microglia phenotype in the absence of any phagocytic stimulus. However, we have checked various signaling pathways controlling microglia polarization like CSF-1R, AKT and CREB and we have not detected any changes in their phosphorylation in the presence of P2X4R agonist/antagonists (data not shown). The other possibility is that P2X4R could control directly phagocytosis and by

doing that, could alter microglia inflammatory response. Indeed, P2X4R activation by IVM induces an increase in myelin endocytosis and degradation in WT mice (see new Fig EV6), which is absent in P2X4^{-/-} mice. Since previous data on literature have observed the expression of P2X4R in lysosome, we have performed new experiments to determine whether P2X4R could directly influence lysosome function. Preliminary data showed that P2X4R stimulation with IVM induces endosome-lysosome fusion and pH acidification (see Fig EV6). These data suggest that P2X4R could potentially modulate myelin phagocytosis directly. This information and hypothesis have been added and discussed in the new version of the manuscript.

1. Please, for the sake of the lay reader, explain what MOG35-55 is, and the rationale for its use, in the Result section.
2. Please, add "+" superscript to CD4 and CD8 in several places throughout the MS.
3. At pg 9, line 19, "is" should be "are".
4. It is kind of unusual that Dr Rassendren is thanked for the gift of the P2X4-KO mice when he is in fact a co-author.
5. A pg 24 (Pain assessment): I do not understand what the reference to "Ugo Basile" means.
6. Sentence at pg 29, lines 9-12, starting with "Naive..." should be amended.
7. Images in Fig. 2B are not really convincing. I urge the Authors to provide better images.

All the minor points have been corrected. Ugo Basile is the company that produces the aesthesiometer. We have improved quality of images in Fig 2B and eliminate Hoechst staining to facilitate cell visualization.

2nd Editorial Decision

25 May 2018

Thank you for the submission of your revised manuscript to EMBO Molecular Medicine. We have now received the enclosed report from the referee asked to re-assess it. As you will see the reviewer is now supportive and I am pleased to inform you that we will be able to accept your manuscript pending final editorial amendments.

Please submit your revised manuscript within two weeks.

I look forward to reading a new revised version of your manuscript as soon as possible.

***** Reviewer's comments *****

Referee #2 (Comments on Novelty/Model System for Author):

This is a well-written and experimentally sound study that might have an important impact in the development of novel treatments for neurodegenerative diseases.

Referee #2 (Remarks for Author):

I think that this MS deserves publication as a full report.

2nd Revision - authors' response

05 June 2018

The authors made the requested editorial changes.

Corresponding Author Name: Maria Domercq
Journal Submitted to: EMBO Molecular Medicine
Manuscript Number: EMM-2017-08743